# A Bayesian model for unsupervised detection of RNA splicing based subtypes in cancers

**David Wang[1,2], Mathieu Quesnel-Vallieres [1,3], San Jewell[1], Moein Elzubeir[1], Kristen Lynch[1,3], Andrei Thomas-Tikhonenko [4,5] & Yoseph Barash [1,6]** ✉

Identification of cancer sub-types is a pivotal step for developing personalized treatment. Specifically, sub-typing based on changes in RNA splicing has been motivated by several recent studies. We thus develop CHESSBOARD, an unsupervised algorithm tailored for RNA splicing data that captures "tiles" in the data, defined by a subset of unique splicing changes in a subset of patients. CHESSBOARD allows for a flexible number of tiles, accounts for uncertainty of splicing quantification, and is able to model missing values as additional signals. We first apply CHESSBOARD to synthetic data to assess its domain specific modeling advantages, followed by analysis of several leukemia datasets. We show detected tiles are reproducible in independent studies, investigate their possible regulatory drivers and probe their relation to known AML mutations. Finally, we demonstrate the potential clinical utility of CHESSBOARD by supplementing mutation based diagnostic assays with discovered splicing profiles to improve drug response correlation.

Analysis of RNA sequencing (RNA-seq) data from large patient cohorts is commonly used to reveal transcriptomic variations that are associated with complex disease. Within the framework of machine learning, such analysis can be framed as either supervised or unsupervised learning tasks. In a supervised setting, the objective is usually to identify transcriptomic variations that act as predictive markers for disease state or are strongly correlated with clinically significant variables[1–3]. Unsupervised analysis typically involves identifying latent substructures in the data which can be used to learn more about disease etiology, such as cancer subtypes[4,5]. One approach to quantify changes in the transcriptome is at the level of alternative splicing (AS). AS is the process by which different segments of pre-mRNA can be removed while others are joined, or spliced, together to form mature mRNA. AS is regulated by intricate interactions between hundreds of RNA binding proteins (RBPs) and is thus highly susceptible to disease-causing disruption, especially in cancer[6–8]. Given the strong association between splice variants and disease, we propose an unsupervised learning algorithm for identifying substructures in a matrix of RNA splicing measurements from cancer patients.

The focus on identifying substructures in RNA splicing cancer data is motivated by several additional observations. First, in cancers such as acute myeloid leukemia (AML), the mutation burden is particularly low such that analyzing genetic mutations alone is insufficient for explaining disruption of key oncogenic pathways[9]. Instead, several works have pointed to splicing aberrations which can severely perturb the function of regulatory proteins involved in apoptosis and cancer suppression[10,11]. Second, many cancers have been shown to be heterogeneous, with patients exhibiting high variability in splicing measurements. While some of this variability is likely due to confounders such as batch effects, recent studies have shown that this variability can result from mutations which seldom appear in a large fraction of the patients[12]. Rather, they are observed in small subsets of patients with both cis acting effects due to mutations at splice sites and trans acting effects due to mutations in splicing regulatory machinery[13]. These observations motivate the derivation of a dedicated method for

[1]Department of Genetics, Perelman School of Medicine, University of Pennsylvania, Philadelphia, PA, USA. [2]Graduate Group in Genomics and Computational Biology, Perelman School of Medicine, University of Pennsylvania, Philadelphia, PA, USA. [3]Department of Biochemistry and Biophysics, Perelman School of Medicine, University of Pennsylvania, Philadelphia, PA, USA. [4]Department of Pathology and Laboratory Medicine, Perelman School of Medicine, University of Pennsylvania, Philadelphia, PA, USA. [5]Division of Cancer Pathobiology, Children's Hospital of Philadelphia, Philadelphia, PA, USA. [6]Department of Computer and Information Sciences, School of Engineering, University of Pennsylvania, Philadelphia, PA, USA. ✉e-mail: yosephb@upenn.edu

identifying "tile" substructures in the RNA splicing data matrix, i.e., subsets of patients that exhibit distinct splicing changes in a subset of genes.

Splicing variations derived from RNA-seq are commonly defined at the level of whole transcripts or at the level of AS "events". Transcript based approaches rely on estimating the abundance of whole isoforms (e.g., RSEM[14], SALMON[15], Kalisto[16]) or relative isoform usage (e.g., MISO[17], BANDITS[18]). In contrast, methods such as MAJIQ[19], SUPPA2[20], rMATS[21], quantify splicing of events, or local splicing variations (LSVs), such as cassette exons. These local splice variations measure splicing as the ratio of RNA segment (e.g., exon) inclusion (defined as percent spliced in or $\Psi \in [0, 1]$) for isoforms that contain the segment vs. isoforms that do not. While quantifying whole isoforms is clearly appealing, we focus here on event based splicing quantification. Some advantages of using AS events include the fact they do not require a-priori assumptions about the underlying isoforms, can handle un-annotated (de novo) isoforms which is crucial in cancer analysis, and can be directly validated with orthogonal methods (e.g., RT-PCR).

Despite the above advantages, there are several modeling challenges associated with unsupervised tile identification using event based quantifications. First, splicing measurements are inherently different from gene expression measurements which are modeled as Gaussian (TPM) or negative binomial (read counts) distributions in many previous works[22–26]. In contrast, $\Psi$ is bounded in the $[0, 1]$ interval, with inclusion levels commonly exhibiting a U shape distribution, favoring either high or low exon inclusion values. Furthermore, $\Psi$ quantifications typically involve only a small subset of reads that span splice junctions, thus accounting for uncertainty of $\Psi$ estimates is important. Finally, when identifying substructures in cancer RNA splicing measurements, it is important to address the inherent heterogeneous nature of the data and natural variations between individuals. Specifically, global patterns across rows ($\Psi$ for specific AS events) or columns (patients) are unlikely. Instead, only a small subset of LSVs may be perturbed in a subset of samples.

Another important challenge we address here, which has implications beyond the analysis of RNA splicing data, is the modeling of missing values. Genomics data often contains missing values that result from technical limitations in sequencing technologies and are assumed to be missing completely at random (MCAR). Under this model, the missingness rate does not depend on observed or unobserved values and can be imputed or ignored[27]. However, in RNA-seq data, the missingness rate is inversely proportional to the sequencing depth where higher read coverage results in a lower probability of missingness. Furthermore, splicing quantifications, unlike expression measurements, cannot be meaningfully imputed since a missing $\Psi$ quantification can be denoted by a 0 or 1 representing alternate junction usage. Thus, naive imputation (e.g., mean) can lead to unlikely intermediate values. This necessitates an alternate model in which values are MNAR (missing not at random). Under this model, the missingness rate depends on observations in the data matrix and external factors such as coverage. In cancer data specifically, values can also be systematically missing due to genetic mutations which could result in a specific junction not being observed (e.g., mutations near splice sites) and should be modeled as a secondary signal.

Here we address the above modeling challenges, by developing CHESSBOARD (Characterizing Heterogeneity of Expression and Splicing by Search for Blocks of Abnormalities and Outliers in RNA Datasets). CHESSBOARD is a Bayesian tile finding algorithm tailored for splicing data with missing values and includes a suite of data processing and visualization tools (Fig. 1). The input consists of a matrix of junction spanning reads counts to account for uncertainty in splicing quantifications (Fig. 1a). The algorithmic task is to identify splicing patterns in the form of tiles in this matrix (Fig. 1b). This is achieved by first employing model based pre-filtering to remove irrelevant LSVs

and reduce and data size (Fig. 1c, left). Next, CHESSBOARD's non-parametric Bayesian tiles model is fit to the data using efficient blocked Gibbs sampling (Fig. 1c center). Finally, posterior summary statistics can be visualized to perform downstream analysis (Fig. 1c right).

We first apply CHESSBOARD to synthetic datasets to show it outperforms several baseline methods and validate the effectiveness of our modeling approach. Next, we show that CHESSBOARD recovers tiles characterized by splicing aberrations which are reproducible in multiple AML patient cohorts. Finally, we show that tiles we discover are correlated with drug responses, pointing to the translational potential of our findings. We also develop GAMBIT (Graphical Annotated Map for Basic Inspection of Tiles), a web-based visualization tool which allows users to visually explore the discovered tile structures and Bayesian output. Both CHESSBOARD and GAMBIT are available as open source tools to facilitate reproducible workflow and analysis.

## Results

### CHESSBOARD robustly models alternative splicing and missing values to discover tile structures in large heterogeneous datasets

To address the challenges of analyzing heterogeneous AS datasets, CHESSBOARD directly models properties of the data that arise from biological and technical processes. Briefly, the model's input data matrix $X$ contains the number of junction spanning reads $x_{ij}$ mapped to the representative (i.e., most variable) splice junction of LSV $j$ in sample $i$ and the total number of reads mapped to the LSV, denoted $\eta_{ij}$ (more input details in Supplementary Note 1.2). Under CHESSBOARD's model, naturally occurring splicing variations in each LSV are captured by a (learned) mixture of a Beta-binomial distribution over each $x_{ij}$ and a binomial distribution (defined by missingness rate $\theta_{j0}$) for having a missing value. This mixture distribution over observed and missing values captures the background. In specific patient subsets however, additional variation or signal in underlying $\Psi$ (captured by a separate Beta-binomial distribution over observed values) or an elevated missingness rate (captured by a separate $\theta_{jl}$) may be observed. Thus, the first part of the CHESSBOARD pipeline is to filter out non-informative splicing events which can be captured well by the background distribution. This is achieved using a parametric bootstrap Kolmogorov-Smirnov test (Supplementary Note 2.1). Then, for the remaining LSVs in the data matrix, CHESSBOARD aims to find latent "tiles" in which multiple LSVs deviate from the background in the same subset of samples. In practice, this means that every sample $i$ belonging to tile $k$ has its (unknown) group indicator variable set $c_i = k$ and every LSVs $j$ belonging to this tile has a matching (also unknown) indicator variable $r_{jk} = 1$. A specific CHESSBOARD model is represented by a learned tile configuration and distribution parameters for all the background and signal groups. Under this Bayesian formulation, every such model can be assigned a posterior probability, and the CHESSBOARD algorithm uses an efficient blocked Gibbs sampling procedure to sample from the posterior distribution over possible models given the observed data matrix $X$. See Methods for a detailed description of the CHESSBOARD model.

In this section we demonstrate the utility of the CHESSBOARD model formulation described above. First, we show that CHESSBOARD accounts for uncertainty in splicing measurements due to low sequencing coverage. Specifically, CHESSBOARD uses a beta binomial distribution which attributes higher variance to LSVs with low coverage, capturing increased uncertainty in their underlying $\Psi$. To assess whether this model is advantageous for estimating variability in splicing data, we simulate $\Psi$ values from a Beta distribution modeling low exon inclusion ($Beta(10, 90)$) and generate reads for each $\Psi$ at various coverage levels from a binomial distribution (Fig. 2a). We compute the empirical variance using MLE (maximum likelihood estimation) under the CHESSBOARD model and two alternative models: a Beta model which also functions on a domain of $[0,1]$ analogous to $\Psi$ values and a

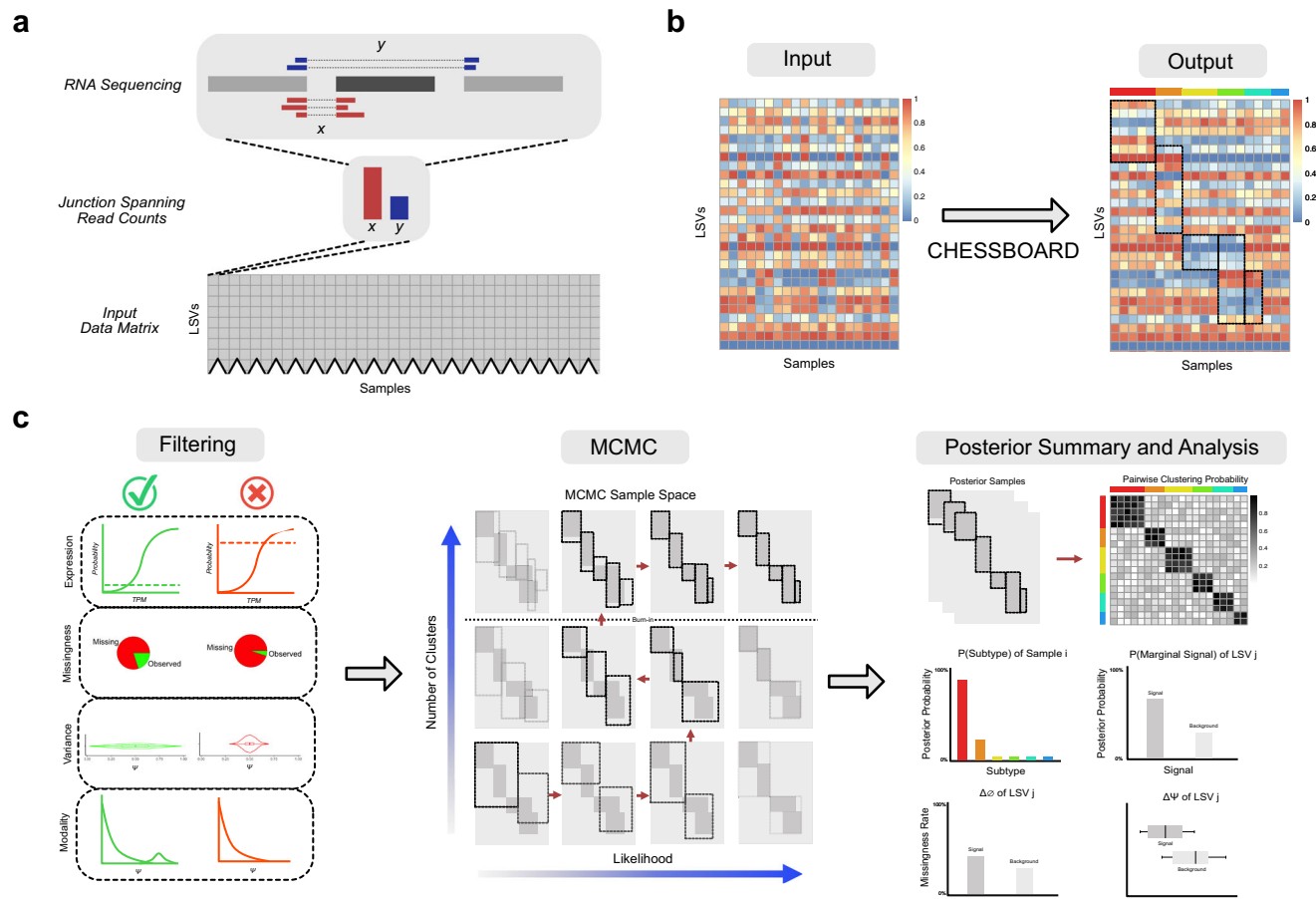

**Fig. 1 | CHESSBOARD Pipeline. a** Input: splice junction read counts (red and blue reads) extracted from patients' RNA sequencing. Each row in the input data matrix is a LSV (e.g., cassette exon shown) and each rubric contains the junction spanning read counts for that LSV in a specific sample. In complex LSV involving more than two junctions, the most variable junction is selected (Methods). **b** Task: CHESS-BOARD's objective is to identify latent tiles in the input matrix. A tile consists of a subset of samples and a subset of LSVs where the Ψ distribution of each LSV for samples within the tile differs from the background distribution. Note that the matrices shown contain Ψ values for visualization purposes but CHESSBOARD acts on the matrix described in (**a**) and it may not be possible to embed each tile as a continuous square in a 2D image as shown here. **c** CHESSBOARD Pipeline: The pipeline includes three steps. Filtering: Lowly expressed genes (lower 5% by default)

and LSVs observed in too few samples (default 20%) are removed, retaining only those exhibiting high Ψ variability between samples and multiple modes in the Ψ value distribution (Methods). MCMC: Blocked Gibbs sampling based on CHESS-BOARD's model and the input data matrix yields posterior samples for potential tile configurations. Intuitively, the algorithm iterates through a chain of solutions that tend toward higher likelihood while varying the number of tiles using the Chinese Restaurant Process (Methods). Analysis: The MC samples are summarized into marginal posterior distributions and possible point estimates for tiles. Tile analysis includes sample assignment to subgroups, LSV assignment to a signal tile, and computation of the ΔΨ and missingness rate associated with a particular LSV in a tile (Methods). Visualization and analysis are conducted using the accompanying visualization package, GAMBIT.

Gaussian model which represents a generic approach. This analysis shows the error in the variance estimation is lowest for the Beta Binomial model and all 3 models converge at about 50 reads. However, in real life datasets the majority of quantifiable LSVs have read coverage significantly lower than 50. For example, in the beatAML and TARGET datasets used in this study (see Fig. 2a as red and blue histograms), 38% and 88% respectively have coverage below this level, indicating a substantial portion of the data benefits from CHESS-BOARD's modeling.

CHESBOARD's model further alleviates the effect of coverage dependent uncertainty in heterogeneous data by sharing information across samples. Specifically, CHESSBOARD uses empirical Bayesian shrinkage to learn group specific priors, taking advantage of samples with higher coverage assigned to the same cluster to improve estimates for samples with lower coverage (Methods). To demonstrate the advantages of CHESSBOARD's modeling approach we generated Ψ values from $Beta(10, 90)$ and read counts from each Ψ at varying levels of coverage as before. The results shown in Fig. 2b demonstrate that indeed there is lower error in Ψ estimates in samples with lower read

counts when estimates are shrunken to the group mean compared to computing $\hat{\Psi}$ without prior information. Furthermore, we show that shrinkage significantly increases correlation of $\hat{\Psi}$ and the true value of Ψ, especially in samples with low coverage. As denoted in Fig. 2c, $\hat{\Psi}$ for darker data points representing low coverage samples is closer to the ground truth with group shrinkage (right) compared to individual quantification (left). Together these experiments show that CHESS-BOARD's generative Beta-Binomial model acting on read counts can substantially improve analysis of splicing data by accounting for uncertainty in the RNA-seq measurements.

Next, we turned to assess CHESSBOARD's missing values modeling. To account for missing values, CHESSBOARD uses a MNAR model where missing values are treated as a secondary signal when the missingness rate of an LSV is much higher than expected under the null missingness rate associated with sequencing limitations. We first replace each unquantifiable splicing event with a missingness indicator. We then estimate priors for the missingness rates using an Empirical Bayes procedure (Methods). During CHESSBOARDs model fit, we obtain posterior estimates for both the background and signal

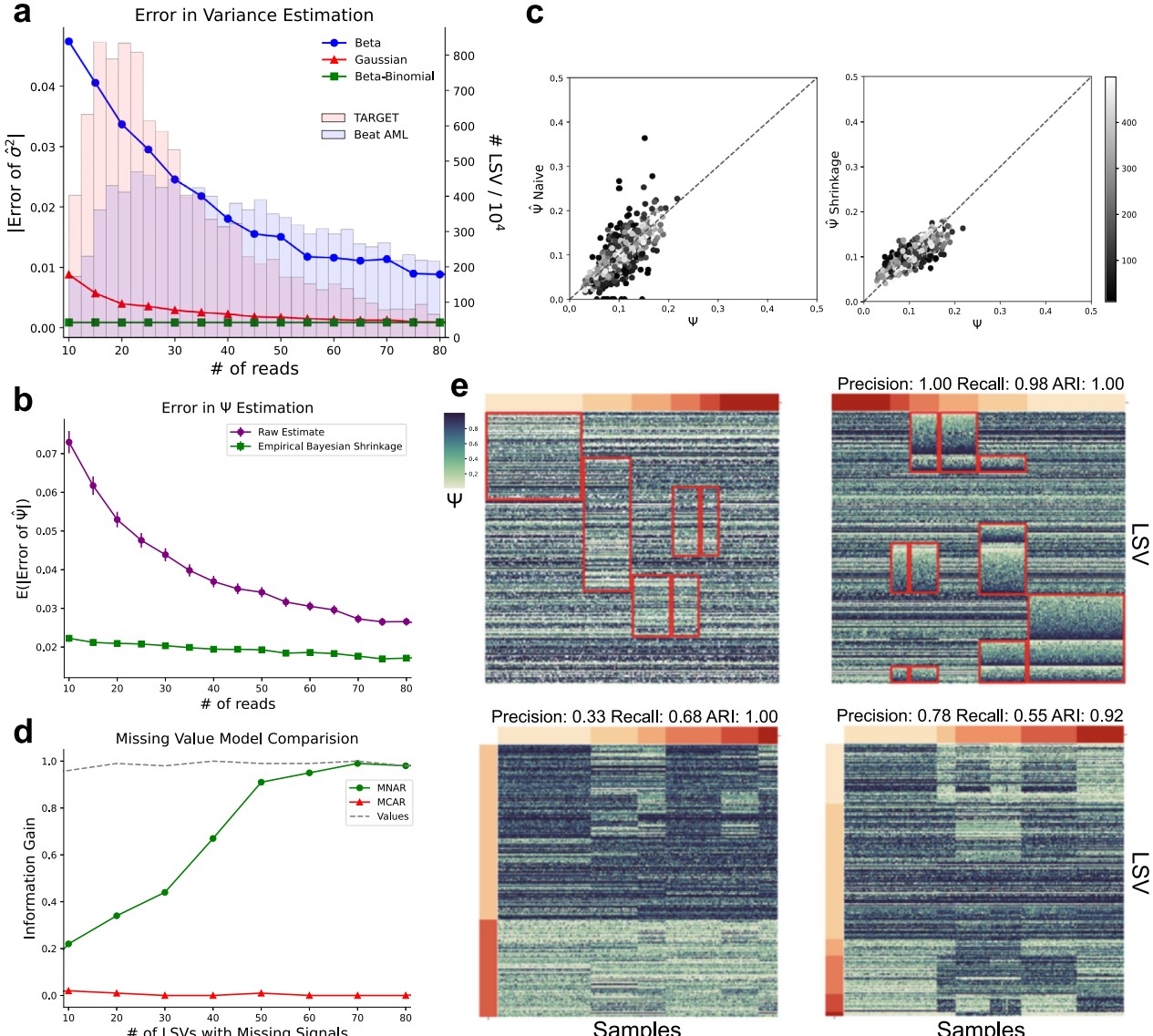

**Fig. 2 | Model Evaluation. a** Error in Ψ variance estimation under Gaussian (red), Beta (blue), and Beta-Binomial (green) models as a function of LSV coverage (*x*-axis). Absolute error in Ψ variance estimates (*y*-axis) is compared to the true variance, assuming a Beta(10,90) distribution. Inset histograms show empirical distributions of LSV coverage in beatAML and TARGET data. **b** Error in Ψ̂ quantification estimates under a naive and empirical shrinkage model as a function of read coverage (*x*-axis, 1000 samples from the same Beta as above for each point). Naive approach uses only read ratios to estimate Ψ̂ while shrinkage model uses the expectation over the posterior for the Beta. Error bars represent the 90% confidence interval for the error in Ψ. **c** Correlation between Ψ and Ψ̂ estimates under a naive (left) and empirical shrinkage model (right). Ψ was sampled as in (**a**) while number of reads *n* represented by the grey scale was sampled randomly from [10,500]. **d** Information gained (Supplementary Note 2.2) from missing signals. Here a background matrix was used, consisting of 100 samples and 100 LSVs with a fixed missingness rate of 10%, into which a signal tile was implanted. The signal tile

consisted of 50 samples and a varying number of LSVs (*x*-axis) with an elevated missingness rate of 60%. The observed values in both tile and background were drawn from the same distribution. Green represents the CHESSBOARD model (MNAR), red represents a missing completely at random (MCAR) version of CHESSBOARD. As a reference, we also plot (gray) the information gain from a similarly sized signal tile where the signal is based on a significantly different Ψ distribution simulated with parameters estimated from real data (Supplementary Note 2.3). Missing signals (green) contribute to an increase in information gain as the number of missing signals increases. **e** Evaluation of CHESSBOARD's (top right) performance on synthetic data, sampled to mimic BeatAML, compared to hierarchical clustering (bottom left) and spectral co-clustering (bottom right). Ψ values are represented as a heatmap, sample groups as colored bars and tiles as red rectangles. Note that tiles may appear permuted. Performance was evaluated using a modified version of recovery relevance score (Supplementary Note 2.2) which is permutation invariant.

---

missingness rate, where the latter can account for other factors such as unobserved values due to mutations (Methods). We show that the MNAR missing value model is effective in identifying tiles containing missing value signals. For comparison, we implemented an alternative version of CHESSBOARD that uses the MCAR model assumption where missing values are integrated out. Both CHESSBOARD and CHESSBOARD-MCAR were then applied to a simulated homogeneous data matrix in which the read counts for each LSV (row) were drawn

from a background distribution with parameters estimated from beatAML and values were missing at a fixed dropout rate of 10%. To these we added a single tile of varying size with a missingness rate of 60%. We then assessed the algorithm's ability to recover this tile by information gain which measures the purity of the clusters (Supplementary Note 2.2). Fig. 2d shows that the MCAR model was unable to identify this tile (information gain close to 0) regardless of tile size, as it relies solely on the observed Ψ values to identify tiles. The MNAR

model is able to effectively recover the tile, but as expected the information gain decreases as the tile size decreases. When the size of the tile increases to 50 LSVs, the information gain reaches a maximum.

After assessing the CHESSBOARD modeling components individually, we turned to assess its ability to recover tiles. For this, we generated synthetic splicing data modeled based on statistics collected from the BeatAML dataset (Supplementary Note 2.3). For comparison, we also ran two commonly used algorithms for biclustering as baselines: single-link hierarchical biclustering (HBC) and spectral co-clustering (SCC)[28]. Since these algorithms lack some of CHESSBOARD's features, both were given data with scalar Ψ and no missing values to fit their input definition. We also ran both with the correct number of clusters given as input. Since CHESSBOARD learns the number of clusters using an infinite mixture modeling approach with a Chinese Restaurant Process (CRP) prior (Methods), we first evaluated the behavior of this feature under various parameters (Supplementary Note 2.4). Then the ability of the algorithms to recover tiles was evaluated using a tile precision ($\tau_{pr}$) and recall ($\tau_{rc}$) statistic adapted from the recovery and relevance score[25] (Supplementary Note 2.2). Intuitively, these scores identify the tile in the test set that maximizes precision or recall with respect to each of the reference tiles, then average the precision or recall across the tiles. In addition, we evaluated sample group clustering using adjusted rand index (ARI) (Supplementary Note 2.2). The results in Fig. 2e show that all algorithms were able to recover sample groups well (ARI > 0.9). This result is to be expected given the strong group signal (number of changing LSV, magnitude of change) in the original data (see BeatAML analysis below) and the fact the baseline algorithms were initialized with the exact cluster number. However, CHESSBOARD significantly outperformed the baseline algorithms in recovering the exact tiles, achieving $\tau_{pr} = 1.00$, $\tau_{rc} = 0.98$ compared to $\tau_{pr} = 0.33$, $\tau_{rc} = 0.68$ for HBC and $\tau_{pr} = 0.78$, $\tau_{rc} = 0.55$ for SCC.

## CHESSBOARD discovers reproducible tiles in AML data which correlate with cancer associated regulators

Having established strong performance of the CHESSBOARD model on synthetic data, we applied it to several primary leukemia sample datasets to discover tiles that correspond to cancer associated regulators. We ran the standard CHESSBOARD pipeline (Supplementary Note 3.1) on the beatAML[12] dataset (samples = 477, LSVs = 2299) (Supplementary Data 1). The algorithm detected a single large tile consisting of 217 samples and 1910 LSVs (Fig. 3a). Confidence in the predicted tile structure was high with most probabilities of sample assignment to the tile cluster and LSV assignment to the signal distribution being close to 1 (Supplementary Fig. 4a). To confirm whether this tile constitutes a real biological signal, we first assessed its reproducibility in Penn HTSC, an independent in-house dataset consisting of 77 adult AML samples. We trained the CHESSBOARD model on a random subset of the beatAML cohort (samples = 400) and used this as a predictive model to predict the tile assignments of the held out beatAML samples (samples = 77) and Penn HTSC (samples = 77) samples (Supplementary Note 3.3)). We used MOCCASIN[29] to account for confounding factors between the datasets. The prediction yielded a similar tile structure in the Penn HTSC dataset (Fig. 3b). Furthermore, the *median*(ΔΨ) (change in Ψ) of LSVs belonging to the tile between the 2 groups in each dataset are highly correlated ($r = 0.779$) suggesting that the splicing perturbations captured by the tile are similar in both datasets (Fig. 3c). Sample likelihoods were also comparable between the held out and external data indicating that the model has similar confidence in the tile structure predictions (Supplementary Fig. 4b).

Having established the reproducibility of the AML splicing tile in two independent cohorts, we then turned to investigate potential mechanisms for formation of this tile. First, we tested whether the identified tile was enriched for differentially spliced junctions that are co-regulated by RNA Binding Proteins (RBPs). Intersecting the tile's differentially spliced junctions with those observed as differentially spliced in ENCODE's RBP knockdown experiments implicated 17 RBPs, all of which were either differentially expressed or spliced between the signal and background patient groups (Supplementary Note 3.4). Put together, all 106 RBPs considered in the analysis affected approximately 11.75% of the junctions in the signal tile. Notably, two RBPs with the most significantly enriched DS junction overlap include *SRSF1* (2.48%) and *U2AF2* (1.54%), both of which have known roles in promoting expression of antiapoptotic isoforms of oncogenes in several hematopoietic maglicanies[30]. Another candidate splicing regulator which appeared to be differentially expressed and spliced between the signal and background groups is *HNRNPC* (2.44%), which has been implicated in AML in a recent study[31]. Next, we analyzed eCLIP data for each RBP to test whether there was evidence for direct RBP binding around the tile's differentially spliced junctions. We observed high binding rates for *SRSF1* and *U2AF2* (>4% of tile junctions). However, the binding rate was lower compared to spliceosome components including *AQR*, *SF3B4*, *PRPF8* and *EFTUD2* which are known to bind spuriously to constitutive splice sites. Surprisingly, almost no binding was observed for *HNRNPC*. For *SRSF1* specifically, there was also significant enrichment of CLIP binding to junctions that were also DS suggesting direct splicing regulation by *SRSF1* (Fig. 3d).

Interestingly, *SRSF1* itself undergoes AS whereby one isoform includes exon 4 for the production of the full protein while the other skips exon 4, resulting in a transcript that contains a premature termination codon that is targeted for nonsense-mediated decay[32]. We thus assessed whether variation in *SRSF1* exon 4 splicing between the 2 clusters corresponds to splicing variations in its known targets. Observed differences in *SRSF1* splicing between the signal and background occurred almost exclusively at exon 4 (Supplementary Fig. 4c). The background cluster had a higher rate of inclusion for exon 4 ($\Psi = 0.759$) compared to the signal cluster ($\Psi = 0.490$) and higher expression of the functional transcript (log2FC = 1.16). This suggests there is higher expression of the productive isoform in the background cluster due to lack of NMD-induced degradation. Over expression of *SRSF1* has been associated with aberrant splicing of several apoptotic factors in cancer[32,33]. We analyzed several cancer-associated genes with experimentally verified splice variations that are affected by *SRSF1* overexpression. Notably, *BIN1* exon 12a inclusion is upregulated in the background and is associated with antiapoptotic processes[33]. Furthermore, exon 3-6 inclusion in *CASP9* is upregulated in the background and is associated with proapoptotic processes[34] (Supplementary Fig. 4c).

## CHESSBOARD offers a gene ranking method that implicates mTORC signaling in identified differentially spliced gene set

In order to more broadly assess whether the identified tiles correspond to known biological functions, we performed a gene ontology analysis of biological processes with the genes harboring LSVs in an extended tile containing all DS LSVs between the two clusters. The analysis shows that genes with differential splicing in the tile are enriched for roles related to general functions commonly found in cancer transcriptomics studies such as gene expression and transcription, RNA processing, and post-translational modifications (Supplementary Fig. 4d). However, we also found that a subset of genes participate in stress-related cellular responses, including regulation of cholesterol/lipid storage and MAPK-signaling (Supplementary Fig. 4d highlighted in red) suggesting that samples from the two clusters exhibit different cellular stress profiles.

To further explore possible tile characterization, we sought to use gene set enrichment analysis (GSEA) to identify similar pathways in the tile gene sets. Since GSEA requires ranking genes

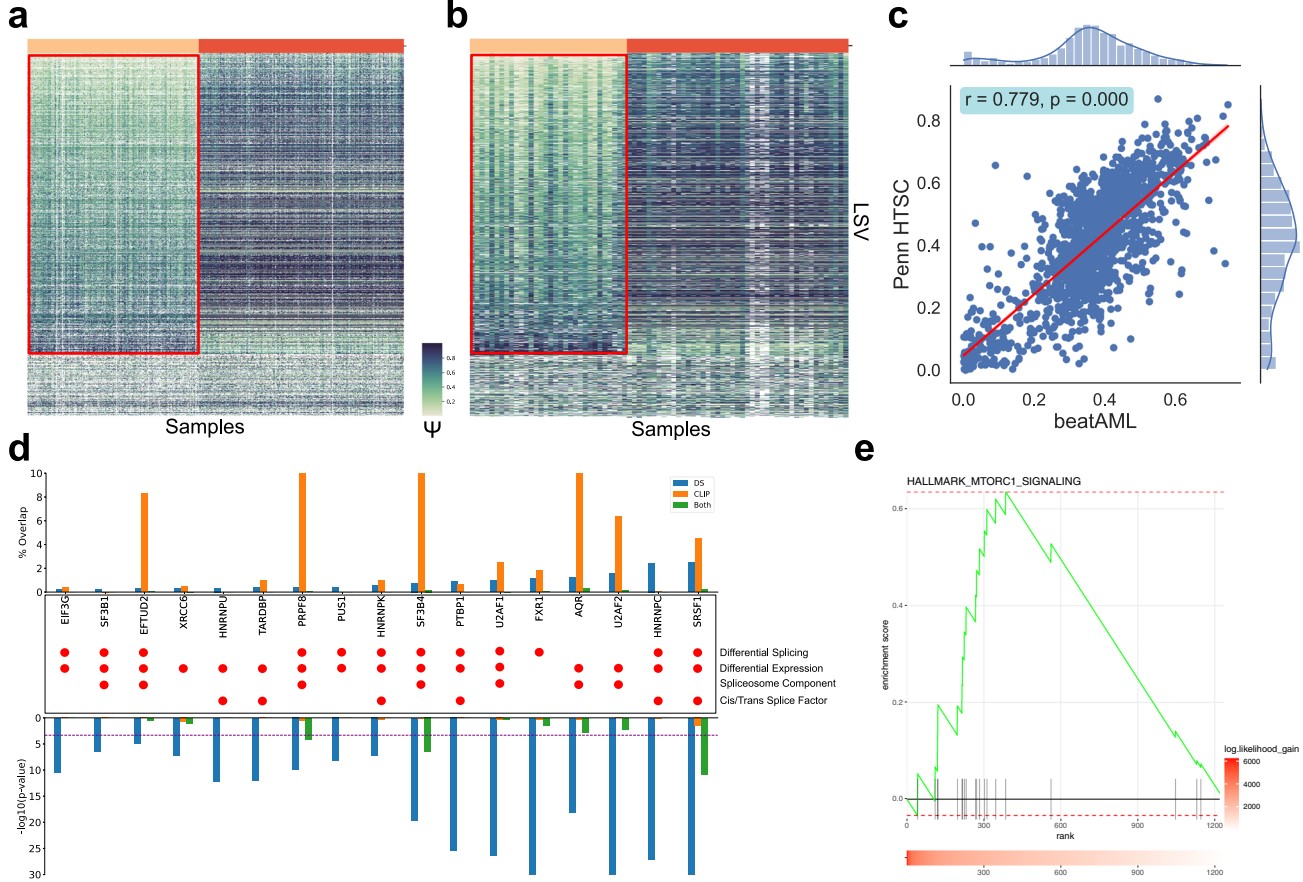

**Fig. 3 | BeatAML Dataset Analysis. a** Heatmap showing the tile discovered by CHESSBOARD on the beatAML dataset (samples = 477, LSVs = 2299). The signal (samples = 217, LSVs = 1910) is outlined in red. Note that although CHESSBOARD was run on junction spanning read rates as input, the heatmap shows Ψ values to facilitate visualization. **b** Heatmap of Ψ values in the Penn HTSC dataset (samples = 77, LSVs = 2299), showing reproducibility of the tiles originally identified by CHESSBOARD in the beatAML dataset. The signal tile (samples = 32, LSVs = 1899) is outlined in red. **c** Correlation between *median*(ΔΨ) in the beatAML and HTSC datasets for the representative junction in each LSVs belonging to the tile. The *median*(ΔΨ) value was computed between the 2 groups discovered by CHESS-BOARD in both datasets. Correlation is measured using Pearson's correlation coefficient (*r*) and the two-sided *p* value is the probability of observing a coefficient > |*r*| under the exact null distribution. **d** ENCODE based analysis of possible tile regulators. Top bar plot shows the percentage of splice junctions (*y*-axis) in the tile

that overlap with splice junctions in one of three categories associated with each RBP/SF (*x*-axis). DS (blue) is the set of junctions that are differentially spliced between knockout and control samples in ENCODE K562 cell lines. CLIP (orange) is the set of junctions that are bound by the RBP in a 250 bp region flanking the junction. The "Both" bar (green) represents junctions in the intersection of DS and CLIP sets. The bottom bar plot shows whether the overlap is significant (bonferroni corrected cutoff) based on a one-sided fisher's exact test for enrichment. The red circles indicate whether the matching RBP/SF is differentially spliced (in at least one junction) and/or differentially expressed between the tiles and whether it is a component of the spliceosome or a cis/trans acting splice factor. **e** mTORC1 GSEA: Enrichment of genes ranked by log(likelihood gain) of LSVs among the HALL-MARK_MTORC1_SIGNALING gene set as performed with GSEA v. 4.1.0 and visualized with the fgsea R package.

within a group we developed a probabilistic ranking method based on the CHESSBOARD model which account for both splicing changes and missingness rates. Specifically, we score each LSV based on the likelihood gain achieved in the learned tile configuration compared to an inverted tile configuration and then used the score for the highest scoring LSV in each gene as input to GSEA (see Supplementary Note 3.5 for details). Indeed, GSEA revealed an enrichment of differentially spliced genes in the tile among the hallmark mTORC1 signaling gene set (Fig. 3e), a signaling pathway centrally involved in stress response[35]. Drawing from experimentally validated interactions extracted from the Ingenuity Pathway Analysis software, we confirmed that several genes that harbor high-ranking LSVs in the tile interact directly with mTORC1 or one of its direct regulators, and that many of these genes activate mTOR signaling, although it is unclear how the splicing variations that we observe might affect the function of the proteins (Supplementary Fig. 4e). Collectively, these

results suggest that the main tile structure CHESSBOARD identified in the BeatAML data represents a highly reproducible and biologically relevant AML subtype.

## CHESSBOARD enables scalable recursive clustering to discover alternate subtype definitions

Although CHESSBOARD was able to successfully discover a tile corresponding to an AML subtype characterized by a specific set of splicing events, other subtype definitions may exist. Alternative tile structures representing these subtypes can emerge when the inclusion of additional features or exclusion of selected features alters the amount of evidence supporting existing tile boundaries. Intuitively, a tile can be interpreted as a collection of correlated transcriptomic signatures that each capture a mis-regulated biological process. For example, the tile discovered in the previous section is partially explained by misregulation of RBPs. Removal of a signal dominated by certain processes can

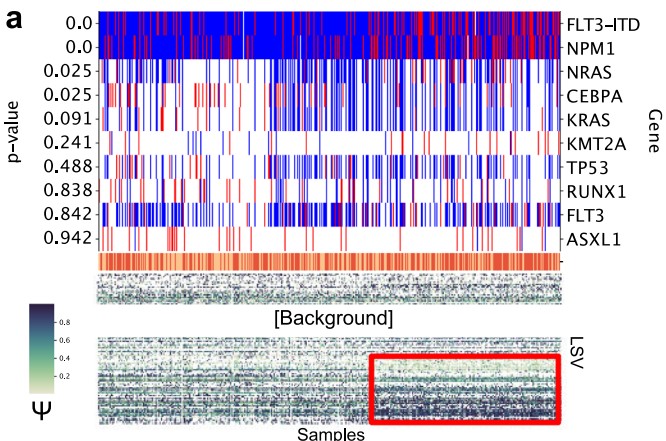

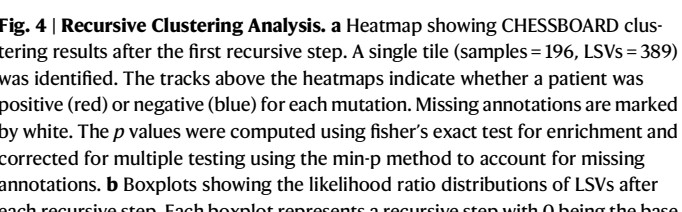

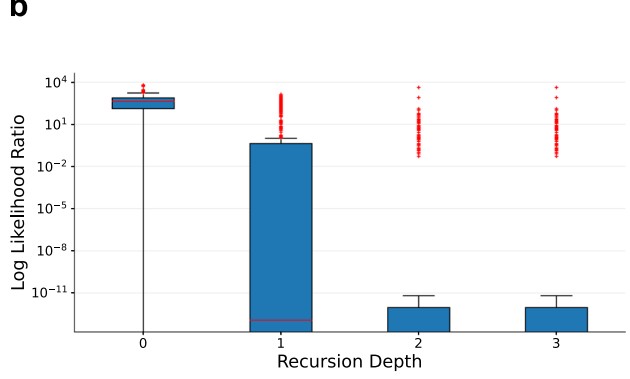

**Fig. 4 | Recursive Clustering Analysis. a** Heatmap showing CHESSBOARD clustering results after the first recursive step. A single tile (samples = 196, LSVs = 389) was identified. The tracks above the heatmaps indicate whether a patient was positive (red) or negative (blue) for each mutation. Missing annotations are marked by white. The $p$ values were computed using fisher's exact test for enrichment and corrected for multiple testing using the min-p method to account for missing annotations. **b** Boxplots showing the likelihood ratio distributions of LSVs after each recursive step. Each boxplot represents a recursive step with 0 being the base case. Within a boxplot, each data point (0:LSVs = 2299, 1:LSVs = 389, 2:LSVs = 330, 3:LSVs = 319) represent the log likelihood ratio of a LSV under the tile model (learned by CHESSBOARD on the original data) and a null model (learned by CHESSBOARD on the original data with tile structures removed by randomly permuting each row of the data matrix). The median is denoted by a red line, the upper and lower quartiles are denoted by the box, the whiskers denote points that lie within 1.5 IQRs of the lower and upper quartile, and observations that fall outside this range are outliers which are independently displayed.

lead to discovery of tiles characterized by misregulation of orthogonal, possibly less pronounced (in terms of number of splicing changes and their magnitude), pathways where other splicing perturbations exist in a different subset of patients. Similarly, LSVs not present in the initial pre-filtered data matrix may provide additional support for such tiles. To address this scenario we developed a recursive clustering solution that naturally fits into CHESSBOARD's probabilistic model and serves as a scalable means to probe the entire transcriptome (Supplementary Note 4.1). In short, our approach iteratively reclusters the LSVs that are not assigned to a tile and after each recursive step, tests for termination by assessing the likelihood ratio of the tile model to a null model in which tile structure is removed. This null model can be interpreted as a distribution over tile structures in datasets where tiles are not expected to occur. Furthermore, the result of each recursive step can be extended to the whole transcriptome using MAJIQ or similar tools for differential splicing analysis between the sample groups identified by CHESSBOARD.

We performed recursive clustering on the beatAML dataset using this approach, treating the result from the previous section as the base case. The first recursive step yielded a smaller signal tile (samples = 196, LSVs = 389) corresponding to a different subset of the patients (ARI = 0.0066 between recursive and base case sample clusters) (Supplementary Data 2). In addition, the new cluster had high correlation with several known AML mutations (Fig. 4a). Specifically, we observed a significant permutation test $p$ value for enrichment of mutations in *FLT3*-ITD ($p < 0.001$), *NPM1* ($p < 0.001$), and *CEBPA* ($p = 0.025$) in patients assigned to the tile cluster (Supplementary Note 4.2). Mutations in these 3 genes are associated with normal karyotype AML which is a known subtype of the disease[36]. We observed a fourth association of the cluster with mutations in *NRAS* ($p = 0.025$), but mutations in this gene were actually depleted in the tile samples. Continued recursive tile discovery showed a sharp decrease in the likelihood ratio (Fig. 4b) indicating there are no more significant tiles in the matrix to discover.

**CHESSBOARD identifies tiles that correlate with drug responses**
To demonstrate the translational utility of conducting a CHESSBOARD analysis, we assessed whether tiles discovered by the algorithm correlate with patient response to therapeutics. We ran the CHESSBOARD

pipeline on the beatAML dataset again but now limited the analysis to only LSVs in 70 AML associated genes (Supplementary Data 3). These genes have been identified as commonly mutated, truncated or translocated in AML patients[36,37] and their mutational status is used by clinicians to decide on drug administration. This targeted tile finding approach based on known gene sets is motivated by several observations. First, we demonstrated above that a transcriptome wide approach can be dominated by signals orthogonal to pathways inhibited by a drug. Second, and as we show below, CHESSBOARD's unsupervised approach can detect splicing signals not directly captured by the mutational landscape in such AML associated genes. Finally, as demonstrated in Rivera et al. 2021[10], clustering splicing changes across those 70 genes gives rise to clear groups and several candidate regulators.

Our splicing analysis of the 70 AML associated genes recovered 2 clusters (Fig. 5a) with resulting patient subgroups similar to the original clustering (ARI = 0.958). This result is notable since the LSV set used for this analysis was significantly different, with shared LSVs constituting only 0.57% of the original LSV set and 14.4% of the current set. This result suggests the splicing changes in AML related genes are part of perturbations to pathways captured in the original, unbiased, LSVs tile finding.

Since the mutational status in many of these AML associated genes is used by clinicians to decide on drug administration we correlated the samples belonging to each tile with drug response measured by area under the $IC_{50}$ curve (AUC). Details about this measurement are discussed in Supplementary Note 5.1. We observed strong correlations between drug response and the tiles, and noticed the tiles included aberrant splicing in many gene targets of the most correlated drugs (Supplementary Data 4). We therefore first tested whether our splicing based patient stratification can serve as good predictors of drug response. Specifically, we computed for each drug the percent of AUC variance that can be explained (Supplementary Note 5.2) by CHESSBOARD's discovered sample groupings compared to that explained by known mutations. However, this analysis conclusively found the variance explained by the patient subgroups to be relatively low, maxing at 6.7% compared to a much higher percentage for known mutations and drug combinations (Fig. 5b). Specifically, the variance explained by *FLT3* mutation is highest for Gilteritinib and *FLT3*-ITD is highest for Sunitinib/ Sorafenib which are known mutation-drug associations.

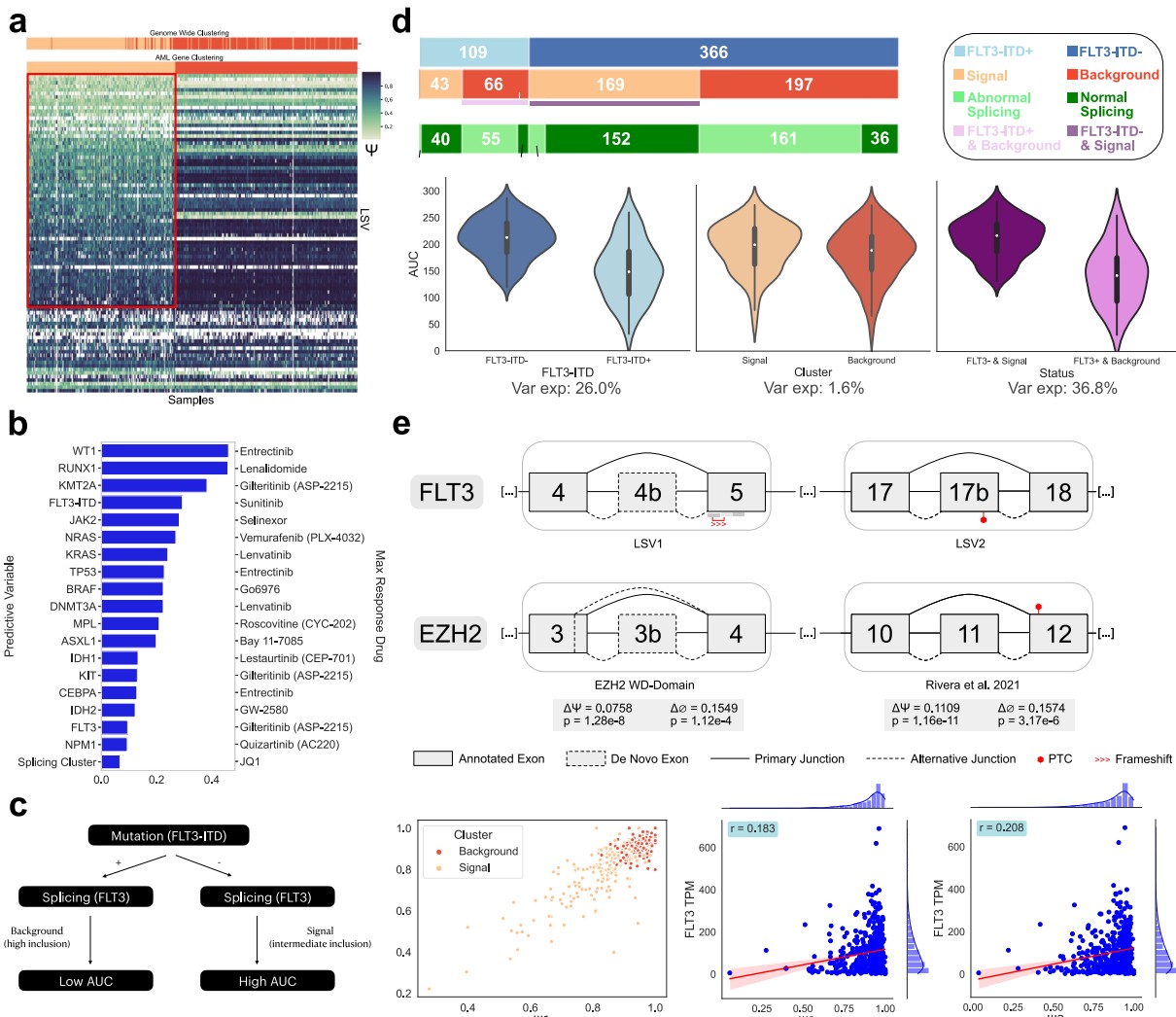

**Fig. 5 | AML Drug Response Analysis. a** Heatmap showing the tile discovered by CHESSBOARD in LSVs from 70 AML related genes (samples = 477, LSVs = 90). The signal (samples = 214, LSVs = 66) is outlined in red. The top "Genome Wide Clustering" track shows sample grouping in Fig. 3a. **b** Barplot showing for each categorical variable (mutation presence or splicing cluster assignment, left) the drug (right) with the maximum AUC variance explained (x-axis) by the corresponding variable. **c** The proposed decision tree for administering Sorafenib based on splicing patterns and mutations. Patients with *FLT3*-ITD- and a signal group splicing pattern exhibit a worse response (high AUC) compared to patients with *FLT3*-ITD+ and a background group splicing pattern (low AUC). **d** Violin plots of AUC values for patients' response to Sorafenib when split according to the groups indicated on the x-axis. When combining both splicing and mutation information using the decision tree in (c), the variance explained increases to 36.8%. The bars at the top indicate the total number of samples that fall into each category. Notably, the groups

exhibiting favorable drug response (*FLT3*-ITD+ & Background) are enriched for abnormal splicing (55/66 patients) while the group with poor response (*FLT3*-ITD– & Signal) are enriched for normal splicing (152/169). Here, abnormal splicing is defined as constitutive expression of the canonical isoform with $\Psi1 > 0.9$ and $\Psi2 > 0.9$. **e** Differential splicing events in *FLT3* and *EZH2* between the subgroups. For *FLT3*, the inclusion of exon 4b in LSV1 and exon 17b in LSV2 results in introduction of a frameshift or PTC respectively. Scatterplot (bottom left) shows correlation between $\Psi$ values for the skipping event in *FLT3* ($\Psi1$ for LSV1, $\Psi2$ for LSV2), while correlation plots (bottom middle and right) show Pearson's correlation between $\Psi$ and *FLT3* expression. The red line indicates the linear regression fit and the band represents the 95% confidence interval. For *EZH2*, the $\Delta\Psi$ values between the clusters for these deleterious events in *EZH2* are low (<0.2), but are part of a change involving a higher rate of missingness in the background cluster (>0.15).

Next, given the possible functional consequences of splicing changes between the identified tiles in many AML associated genes, we hypothesized that our splicing based patient grouping could improve clinical decisions that are based on mutation analysis alone. Notably, the added value of splicing changes to AML classification has been shown recently for *FLT3*-ITD and *NPM1* for FAB classification AML genes[38] as well as RUNX1 and SF3B1[39]. To assess usefulness of combining splicing changes and mutations to predict drug response, we prioritize *FLT3*-ITD and Sorafenib and *NPM1* and Venetoclax due to reliable mutation calls and their prominent role in AML clinical diagnosis. Specifically, we developed a simple decision tree (Fig. 5c)

combining both *FLT3*-ITD and patient subgroups which increased the AUC variance explained by *FLT3*-ITD for Sorafenib from 26.0 to 36.8% (Fig. 5d). We also looked at median change in AUC and $IC_{50}$ fold change (FC) to confirm that the effect size differences between the groups is biologically meaningful. Accordingly, using *FLT3*-ITD alone had median($\Delta$AUC) = 64.36 and Log3FC = 2.09, while the combined classification had a median($\Delta$AUC) = 76.29 and Log3FC = 2.73 ($p = 0.034$ by permutation test, see Supplementary Note 5.3). Similarly for *NPM1*, using the *NPM1* mutation status alone had a median($\Delta$AUC) = 23.34 and Log3FC = 0.86, while the combined classification had a median($\Delta$AUC) = 57.91 and Log3FC = 2.65 ($p = 0.048$).

## CHESSBOARD's identified tiles include splicing changes in AML drugs' target genes

To assess potential mechanisms by which CHESSBOARD tiles explain drug response as described above, we looked for specific targets of Sorafenib in the tile that were differentially spliced between the 2 groups. Sorafenib is commonly used as a treatment for AML patients with a *FLT3*-ITD mutation and functions as a tyrosine kinase inhibitor with high specificity for *FLT3*. At a molecular level, *FLT3*-ITD result in constitutive activation of receptors that lead to downstream activation of PI3K/AKT/mTOR, Ras/Raf/MEK/ERK, and JAK/STAT pathways. This activation in turn results in enhanced proliferation and reduced apoptosis of the myeloblasts, which contribute to leukemogenesis[40]. Inline with this known mechanism, we observed multiple LSVs in *FLT3* that were differentially spliced between the patients' subgroups. We therefore analyzed several specific splicing events in *FLT3* to determine if there was enrichment of splice isoforms in the signal that may lead to reduced transcript viability and thus higher sensitivity. Indeed, for *FLT3* we identified two differential splicing events involving skipping of exon 4b ($p = 1.13e{-43}$, $\Delta\Psi = 0.110$) and exon 17b ($p = 4.34e{-33}$, $\Delta\Psi = 0.115$) that are highly correlated ($r = 0.858$) and have a higher skipping rate in the background. Notably, exon 4b has not been previously reported (de novo exon and junctions) and both events have not been previously reported with respect to AML to the best of our knowledge. Skipping both of these exons results in the functional canonical isoform of *FLT3* which was correlated with an increase in expression of *FLT3* (Fig. 5e). In contrast, inclusion of this exon introduces a PTC or frameshift in the alternate isoform.

Taken together, the above analysis suggests that there is overexpression of *FLT3* in the background cluster due to constitutive expression of canonical *FLT3* and failure of regulatory systems to induce NMD. Noting that Sorafenib is a tyrosine kinase inhibitor, which includes *FLT3*, an increase in the concentration of its target would therefore be expected to increase drug sensitivity, as the splicing changes we detected could mimic the gain of function effect of *FLT3*-ITD. Inline with this mechanistic hypothesis, we find that when combining the splicing signals and *FLT3*-ITD status, the group of patients that were *FLT3*-ITD negative and assigned to the signal group had much worse responses than patients that were *FLT3*-ITD positive and assigned the background tile (Fig. 5d). Indeed, this latter group of patients has significant enrichment of patients (55/66 patients, $p = 8.95e{-8}$) with constitutive *FLT3* canonical isoform expression, defined as inclusion of both events being > 0.9. In contrast, the patients that were *FLT3*-ITD- and in the signal group showed enrichment for intermediate canonical isoform levels (152/159 patients, $p = 4.59e{-22}$).

Finally, in another investigation of tile associated splicing changes in AML genes we observed high enrichment of missing values in the background cluster for *EZH2*. The change in inclusion levels was not particularly large, yet there was over a 15% increased missingness rate of two *EZH2* LSV in the background cluster (Fig. 5e). One of these events corresponds to an event recently reported by Rivera et al. 2021[10] and validated to introduce a PTC that induces NMD and results in reduced protein levels. The other splicing change we identified introduces an unannotated exon into the highly conserved WD domain of the protein. This suggests there was rapid degradation of the transcript making it more difficult to sequence which in turn resulted in elevated missing values. In summary, our analysis of CHESSBOARD's tile with respect to drug response indicated that the RNA splicing tiles correlate with AML specific drug responses and offer insights into potential underlying mechanisms captured by both changes of $\Psi$ and missing values.

## CHESSBOARD finds more complex tile structures in other leukemia datasets

While our analysis focused on adult AML, we also applied CHESSBOARD to several other datasets and disease to demonstrate CHESSBOARD's general utility for splicing pattern discovery. First we applied CHESSBOARD to a joint dataset (samples = 1089, LSV = 2965) consisting of TARGET pediatric AML and beatAML samples (Supplementary Data 5). Studies have show that there are many genetic differences between pediatric and adult AML[41]. However the mutation burden in pediatric AML is lower suggesting that alternative disease-causing modalities should be investigated. Specifically, LSVs that are included in tiles that are enriched for samples of a single disease type can be used to distinguish the diseases at the transcriptomic level. On the other hand, LSVs which appear in tiles with mixed sample composition represent splicing variations that are shared between diseases. CHESSBOARD discovered 5 clusters in this dataset. Notably, tiles segregate by disease with C1, C2, and C4 representing pediatric AML and C3 and C5 representing adult AML (Fig. 6a). However a subset of LSVs are unique to adult (green) and pediatric (blue) AMLs respectively. Other LSVs are either shared between subtypes of each disease type (yellow) or unique to only a single subtype of a disease (purple). Many of these splice variations occur in genes that are commonly differentially mutated between pediatric and adult disease types[42,43].

Next we applied CHESSBOARD to TARGET B-ALL (B-cell Acute Lymphoblastic Leukemia) data (samples = 517, LSVs = 1562), a markedly different type of leukemia characterized by proliferation of lymphoid blasts in the bone marrow (Supplementary Data 6). We recovered five clusters with a distinctively more complex tile structure compared to the result on the beatAML dataset (Fig. 6b). Of note, one identified subgroup was enriched for patients which are *RUNX1-ETV6* fusion negative who also have high relapse rates. This mutation is often used as a positive prognostic marker which suggests the splicing signature associated with this tile can be used in a similar manner[44].

## Discussion

There is increasing evidence for the pathogenicity of splicing aberrations in heterogeneous cancers such as AML and B-ALL[10,11,45,46], pointing to a need for methods dedicated to unsupervised discovery of splicing based disease subtypes. Here, we develop CHESSBOARD, a method which offers several contributions to the densely populated area of clustering and missing value modeling. Specifically, previous works on tile finding and biclustering approaches were either not domain specific[23,24] or tailored for other data modalities such as gene expression and genetic mutations[22,25,26]. Consequently, these algorithms do not consider crucial characteristics of heterogeneous splicing cancer data such as the uncertainty in splicing quantifications and missing values. We demonstrate here using both synthetic and real data, the usefulness of modeling these data characteristics. Furthermore, CHESSBOARD's MNAR model could also be applicable in domains well beyond RNA splicing or even clustering, for example in algorithms for dimensionality reduction such as sparse probabilistic PCA or factor analysis[47,48].

Beyond the CHESSBOARD model, we also implement several additional algorithms and tools to enable more extensive exploration of the data. First, we developed a pre-filtering and recursive clustering method to facilitate analysis of the entire transcriptome. We then used the recursive clustering to discover alternate AML subgroups definitions which strongly correlated with mutations in key AML genes. Second, we implement a LSV ranking system to enable prioritization of driver genes for use in downstream analysis like GSEA. This system is unique in that we can rank LSVs based on differences in $\Psi$ distribution and enrichment of missingess value signals. Finally we implement the CHESSBOARD algorithm and all analytical tools in a Python package. The package is accompanied by an online interactive visualization tool called GAMBIT that enables users to manually inspect the LSVs and samples contained in each tile.

While we applied CHESSBOARD to several leukemia datasets, we focused on beatAML as it offered both a large set of samples and drug response measurements. In beatAML, we found a single strong "signal"

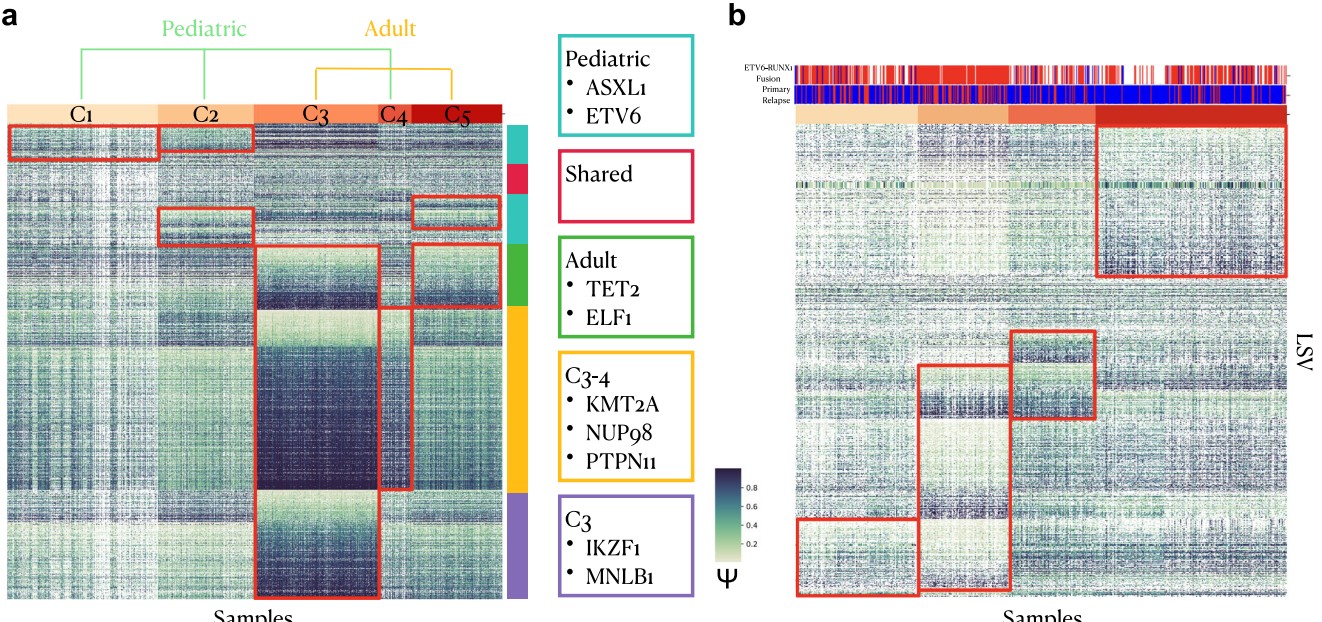

**Fig. 6 | Pediatric AML and B-ALL Analysis. a** Heatmap showing the tiles discovered by CHESSBOARD when applied the joint beatAML and TARGET pediatric AML dataset (samples = 1089, LSV = 2965). Tiles are outlined in red. Track on the y-axis groups the LSVs into groups defined as: unique to pediatric (blue), shared between diseases (red), unique to adult (green), unique to one subtype in each disease (yellow), unique to only 1 disease disease and subtype (purple). **b** Heatmap showing the tiles discovered by CHESSBOARD when applied the TARGET B-ALL dataset (samples = 517,LSVs = 1562). Tiles are outlined in red. The top track indicates whether the patient is positive (blue) or negative (red) of *RUNX1-ETV6* fusion. The second track indicates where the sample is primary (blue) or relapse (red).

tile that divided the dataset into two main subgroups of patients which were highly reproducible in an independent dataset. Investigating possible splicing factors which may form these tiles, we find that *SRSF1* is a key regulatory factor and affects the splicing of 2.49% of the junctions in the tile through direct binding. However it is important to note that taken together all of these RBPs can still only explain 11.75% of the splicing variations in the observed tiles. This arguably low fraction could be due to a myriad of reasons, including the difference between ENCODE's cell lines and tumor specimens, the limited number of RBPs served by ENCODE, and inherent noise in the CLIP and KD assays.

When we investigated possible functional consequences of the two BeatAML subgroups, we found that the genes containing events differentiating the groups were enriched for genes in the mTORC1 pathway. As mTOR is frequently activated in cancer in a manner that affects drug susceptibility[49,50], our clusters might reflect variations in cellular metabolism that could alter drug susceptibilities in AML samples. Furthermore, we suspect that the signal tile corresponds to a subtype of AML that may be less adverse (see Supplemental Note 7.1 and Supplementary Fig. 7). Inline with this hypothesis, we find that the background tile was characterized by *SRSF1* misregulation which affects several oncogenes including *BIN1* and *CASP9* in the tile.

We demonstrated the utility of CHESSBOARD's recursive clustering by detecting an alternative tile in the BeatAML data which correlated with *FLT3*-ITD, *NPM1*, and *CEBPA* mutations, defined together as normal karyotype AML[36]. The discovery of this known subtype points to the power of recursive clustering. By removing the dominant signal driven by splicing variations caused by misregulation of RBPs/SFs, we enabled further discovery of an alternate tile structure associated with a different AML subtype characterized by weaker splicing signals but a strong mutation signature. We then demonstrated the clinical utility of CHESSBOARD by analyzing correlation of tiles with drug response data. Notably, we found that while mutations were better predictors of drug response than splicing signals, combining the two yielded a better prediction overall, specifically for *FLT3*-ITD and Sorafenib and *NPM1* and Venetoclax. An interesting hypothesis related to these results is that Sorafenib sensitivity may have been reduced by

enrichment of the PI3K/mTOR pathway in the signal group as suggested by previous work[51]. Indeed, such a connection between Sorafenib and the mTOR pathway has also been observed in hepatocellular carcinoma where treatment with Sorafenib in patients with increased PI3K/mTOR pathway activity results in reduced relapse rates[52–54]. A similar effect has recently been observed in AML patients too[55].

There are several limitations in this study which are important to highlight. Specifically, the narrow $IC_{50}$ concentration ranges used in the beatAML experiments limited fitting of sensitivity curves and thus we had to use AUC as a proxy for sensitivity. Furthermore, despite the many advantages of the CHESSBOARD model, we make several modeling assumptions that could be improved upon. For example, CHESSBOARD assumes there is only a single signal distribution. In many scenarios, there can be multiple sources of heterogeneity that lead to signal distributions that are a mixture of Beta distributions. We note though that in practice, given the noisy nature of splicing and its quantification from limited read counts, we did not find many clear LSV cases in the data used here that would justify the additional complexity beyond a two component mixture model of signal vs. background.

In summary, we developed CHESSBOARD, the first RNA splicing tailored algorithm for signal detection in heterogeneous RNA-seq datasets. We showed its applicability on several leukemia datasets, connecting the splicing tiles discovered to potential regulators, drug response, and known pathways. Although we present a model of splicing, CHESSBOARD can be easily adapted for alternate datatypes such as expression and multi-omics data integration using a multiview model[56]. We also hope the research community will take advantage of the open source code and apply CHESSBOARD to many other analysis tasks in large, heterogeneous cancer datasets, pushing further our understanding of the role of splicing in complex disease.

## Methods
### Filtering
To enable analysis of large datasets, CHESSBOARD uses a pre-filtering pipeline to select LSVs of interest followed by a recursive clustering

procedure. This two-step process allows the algorithm to analyze the most potentially interesting splicing events at a high resolution by removing noisy events that could potentially confound true signals. The filtering pipeline is detailed below:

- **Remove LSVs that correspond to lowly expressed genes**. We quantified gene expression using Salmon and aggregated transcript level quantifications into gene level quantifications by summing the TPMs. Any LSV corresponding to a gene with TPMs in the lowest 5th percentile was removed from further analysis.
- **Remove LSVs with high missingness rates**. A LSV is considered quantifiable if at least 10 reads are observed as being mapped to its splice junctions. Any LSV that is not quantifiable in more than 80% of the samples is removed. Note that we allow for such a high missingness rate because the algorithm is designed to handle missing values.
- **Select highly variable LSVs**. For each LSV j in sample i, we compute the variance across all samples $\sigma_j^2 = \sum_i^N \frac{(\Psi_{ij} - \mu_j)^2}{N}$. We construct the empirical CDF of variances and choose a cutoff based on where the graph plateaus. This procedure selects for approximated 1500–2500 events in our datasets.
- **Select for LSVs with a bimodal $\Psi$ distribution**. Intuitively, a mode that tends toward 0 or 1 with low variance is likely to represent a background distribution since most splicing events favor high or low inclusion. A mode with high variance favoring intermediate values is likely to represent an interesting biological signal that could explain disease state. To select for bimodal LSVs, we use the parametric-bootstrap Kolmogorov-Smirnov test. Under this test, the null hypothesis $H_0$ is the data was drawn from a single component beta distribution while the alternate hypothesis $H_1$ is the data was drawn from multiple beta distributions. The steps for the test are as follows:

  - For each LSV j, fit a beta distribution to the observed $\Psi$ values by obtaining the maximum likelihood estimates of $\alpha$ and $\beta$. Since there is no closed for solution for the MLE, we optimize it numerically.
  - Obtain the observed test statistic, the Kolmogorov-Smirnov $D$, using a 1 sample KS test with the observed data and the CDF of $Beta(\hat{\alpha}, \hat{\beta})$
  - Given $\hat{\alpha}$ and $\hat{\beta}$, simulate $B$ bootstrapped datasets.
  - For each bootstrapped dataset, estimate $\hat{\alpha}_b$ and $\hat{\beta}_b$ and compute $D_b$.
  - Compute the empirical $p$ value of the test as the fraction of boostrapped test statics that are greater than the observed test statistic.

We then select all LSVs with $p < 0.05$. This can be interpreted as selecting LSVs that are multimodal with a 5% chance of being a false positive. If a lower proportion of false positives is desired, one could correct the false discovery rate using a procedure such as Benjamini–Hochberg.

## Modeling observed splicing events

Consider a data matrix $X_{n \times m}$ with $n$ columns representing patient samples and $m$ rows representing AS events or LSVs. For a given sample $i$ and LSV $j$, $x_{ij}$ contains the number of junction spanning reads that are mapped to a splice junction of interest while $\eta_{ij}$ denotes the total number of reads mapped to all junctions in the LSV (e.g two alternative 5′ splice sites of an exon). Under CHESSBOARD's formulation, each sample has an unobserved label $\{c_1, c_2, ..., c_n\}$ which assigns it to a patients' group or type $k \in \mathcal{Z}^+$. Each such group $k$ is defined by a vector $\boldsymbol{r}_k \in \{0,1\}^m$ where $m$ is the dimension of the vector. The assignment $r_{jk} = 0$ indicates LSV $j$ is not part of the unique pattern of group $k$ such that observed inclusion levels for this LSV in samples that do not belong to group $k$ follow some (learned) background $\Psi$ distribution. In contrast, $r_{jk} = 1$ indicates an abnormal splicing signal in LSV

$j$ across all samples belonging to group $k$. We thus formulate the generative process for each observed $\Psi$ entry of the data matrix as

$$
\begin{aligned}
x_{ij} &\sim Binomial(\eta_{ij}, \Psi_{ij}) \\
\Psi_{ij}|c_i = k, r_{jk} &\sim r_{jk} Beta(\mu_{j1}, \kappa_1) + (1 - r_{jk}) Beta(\mu_{j0}, \kappa_0) \\
\mu_{j0} &\sim Beta(\alpha_0, \beta_0) \\
\mu_{j1} &\sim Beta(\alpha_1, \beta_1) \\
\boldsymbol{r}_k &\sim Bernoulli(\boldsymbol{\delta}) \\
\sum_k r_{jk} &\sim Exp(\lambda) \\
c_i &\sim Categorical(\phi) \\
\phi &\sim Dirichlet(\alpha_o / K)
\end{aligned} \tag{1}
$$

A plate visualization of this model is shown in Supplementary Fig. 8. A table documenting all variables is given in Supplementary Note 6.1. Under this model, the read rate of sample $i$ in LSV $j$ follows a Binomial distribution with a Beta mixture prior over the level of inclusion $\Psi_{ij}$. This Beta-Binomial model naturally handles uncertainty in $\Psi$ estimates since observations with low read counts will have higher variance. If observation $x_{ij}$ is assigned to the signal in group $k$ as denoted by $c_i = k, r_{jk} = 1$, its likelihood is evaluated using the signal prior distribution $Beta(\mu_{j1}, \kappa_1)$. Likewise, the observation is evaluated using the background prior distribution $Beta(\mu_{j0}, \kappa_0)$ when $r_{jk} = 0$. Notice that we reparameterize the Beta in terms of mean and variance using $\mu_j = \alpha_j / (\alpha_j + \beta_j)$ and $\kappa_j = (\mu_j(1 - \mu_j))/\sigma_j^2$ where the concentration $\kappa$ is inversely proportional to variance. This reparameterization enables the use of a Beta hyperprior over the mean of each mixture component to capture known biological behavior of AS. Specifically, normal splicing dynamics have a propensity toward high or low inclusion levels which can be modeled using the Jeffery prior $\alpha_0 = 0.5, \beta_0 = 0.5$ and high concentration $\kappa$. Intermediate levels of inclusion modeled by a distribution with a long tail generally indicate aberrant splicing and can be modeled as a Beta distribution with $\alpha_1 = \beta_1$ and low concentration. To control the number of tiles, we impose a L1 penalty with hyperparameter $\lambda$ to induce sparsity in the number of groups for which LSV $j$ is assigned to the signal distribution. Specifically, $\sum_k r_{jk} \sim Exp(\lambda)$.

In most biological contexts, the number of tiles is unknown a priori. This can be modeled as an infinite mixture of groups using a Dirichlet Process prior with Bernoulli base distribution

$$
\begin{aligned}
\Psi_i|c_i = k, \boldsymbol{r} &\sim f(\boldsymbol{r}_k) \\
\boldsymbol{r}_k &\sim G \\
G &\sim DP(G_0, \alpha_o) \\
G_0 &\equiv Bernoulli(\delta)^m \\
c_i &\sim CRP(\alpha_o)
\end{aligned} \tag{2}
$$

where $G_0$ is $Bernoulli(\delta)^m$ and $\alpha_0$ is the concentration parameter. A larger value of $\alpha$ will result in the discovery of more sample groups. In taking the limit of $k$, the distribution of $c_i$ can be interpreted as a distribution of partitions of natural numbers which is usually formulated as the CRP or stick breaking process.

## Modeling missing values

In gene expression data, missing values typically arise due to low sequencing coverage which results in some transcripts lacking any observable reads even if they are expressed. However in splicing data, a lack of junction spanning reads mapped to a transcript that is expressed indicates inclusion of an alternative exon. Many algorithms handle missing values under the MCAR model of missingness by integrating missing values out of the model[27]. Under this model, the missingness rate of a feature does not depend on any observed or unobserved values. However, this

is generally only a valid assumption in scenarios when the data generating instrument malfunctions such as a defective microarray probe. The missingness rate of transcriptomic quantifications from RNA-seq is proportional to sequencing depth thus a model in which values are MNAR (missing not at random) will yield better estimates for the missingness rate. Under this model, the missingness rate of features depends on observations in the data matrix and external factors. In cancer data specifically, values can also be systematically missing due to genetic mutations resulting in no reads being mappable to the splice junction. This can occur when a mutation near a splice site reduces junction usage due to changes in splice factor binding or when the mutation introduces a PTC into an exon resulting in rapid degradation of the transcript due to NMD. Thus it becomes necessary to treat missing values as a secondary signal when the missingness rate of an LSV is much higher than expected under the null missingness rate associated with sequencing limitations. To handle missing values under the MNAR model and detect missing value signals, CHESSBOARD identifies unquantifiable LSVs with $\eta_{ij} < 10$ and replaces $x_{ij}$ with indicator $\omega_{ij} = 1$. The indicator is modeled as

$$
\begin{aligned}
\omega_{ij}|c_i = k, r_{jk} &\sim r_{jk} Bernoulli(\theta_{j1}) + (1 - r_{jk}) Bernoulli(\theta_{j0}) \\
\theta_{j0} &\sim Beta(a_{j0}, b_{j0}) \\
\theta_{j1} &\sim Beta(a_{j1}, b_{j1})
\end{aligned} \tag{3}
$$

where $\theta_0$ is the background missingness rate that represents values that are missing due to techinical factors such as coverage and $\theta_1$ represents the signal missingness rate that is expected to be higher and represents values that are missing due to mutations. The priors can be estimated empirically. Specifically, we estimate the background priors by fitting the following Beta-Binomial regression model.

$$
\begin{aligned}
\upsilon_j &\sim BetaBinomial(n, \mu_j\Phi, (1 - \mu_j)\Phi) \\
logit(\mu_j) &= \beta_0 + \beta_1\chi_j
\end{aligned} \tag{4}
$$

Here, $\chi_j = median(\{\eta_{1j}, \eta_{2j}, ..., \eta_{nj}\})$ is the median number of reads $\eta$ that are mapped to LSV $j$ across the $n$ samples. $\upsilon_j$ is the number of samples with missing observations for LSV $j$. The fitted model returns MLE estimates for the coefficients $\beta_0$ and $\beta_1$ and the dispersion $\Phi$. This trained model can then be used to estimate background priors $\alpha_{j0} = \mu_j\Phi$ and $\beta_{j0} = (1 - \mu_j)\Phi$ by predicting $\mu_j$ from the median read depths for each LSV $j$ in the cancer data to be analyzed. In this study the above model was fitted to whole blood samples from GTEX V8. Users can of course fit the model to more relevant healthy tissue samples for their specific cancer of interest. Generally, we expat that in healthy control samples the missingness rate will be inversely proportional to sequencing depth (MNAR) but these missing values would not represent signals caused by cancer. In a similar way, we use the same procedure over the training data (beatAML or TARGET) to get estimates for the matching signal prior $\alpha_{j1}$ and $\beta_{j1}$.

**Posterior sampling**

The entire joint likelihood of the CHESSBOARD model is given by:

$$
\begin{aligned}
&P(\boldsymbol{x}, \boldsymbol{\omega}|\boldsymbol{\eta}, \boldsymbol{c}, \boldsymbol{r}, \boldsymbol{\Psi}, \boldsymbol{\theta}, \boldsymbol{\mu}, \boldsymbol{\kappa}, \boldsymbol{\alpha}, \boldsymbol{\beta}, \lambda) \propto \\
&\prod_{(i,j) \in \{i,j | \forall \omega_{ij} = 0\}} P(x_{ij}|\eta_{ij}, \Psi_{ij}) P(\Psi_{ij}|c_i, r_{jc_i}, \mu_{j1}, \mu_{j0}, \kappa_1, \kappa_0) P(\mu_{j1}|\alpha_1, \beta_1) P(\mu_{j0}|\alpha_0, \beta_0) \\
&\prod_{i=1}^{n} \prod_{j=1}^{m} P(\omega_{ij}|c_i, r_{jc_i}, \theta_{j0}, \theta_{j1}) P(\theta_{j0}|, a_{j0}, b_{j0}) P(\theta_{j1}|, a_{j1}, b_{j01}) P\left(\sum_{k \in \{c\}} r_{jk}|\lambda\right)
\end{aligned} \tag{5}
$$

where a bold variable indicates a vector containing the variables across all possible indices. To sample from the model's posterior, we develop

an efficient blocked Gibbs sampling scheme which we will use to sample from each conditional posterior. The full conditional posterior of $c_i$ is denoted by

$$
\begin{aligned}
&P(c_i = k|\boldsymbol{\eta}, \boldsymbol{r}, \boldsymbol{\Psi}, \boldsymbol{\theta}, \boldsymbol{\mu}, \boldsymbol{\kappa}, \boldsymbol{x}, \boldsymbol{\omega}, \boldsymbol{\alpha}, \boldsymbol{\beta}, \lambda) \propto \\
&\prod_{j \in \{j | \forall \omega_{ij} = 0\}} P(x_{ij}|\Psi_{ij}) P(\Psi_{ij}|c_i = k, r_{jk}, \mu_{j1}, \mu_{j0}, \kappa_1, \kappa_0) P(\mu_{j1}|\alpha_1, \beta_1) P(\mu_{j0}|\alpha_0, \beta_0) \\
&\prod_{j}^{m} P(\omega_{ij}|c_i = k, r_{jk}, \theta_{j0}, \theta_{j1}) P(\theta_{j0}|, a_{j0}, b_{j0}) P(\theta_{j1}|, a_{j1}, b_{j01}) P\left(\sum_{k \in \{c\}} r_{jk}|\lambda\right) P(c_i = k)
\end{aligned} \tag{6}
$$

Due to beta-binomial conjugacy, we can integrate out $\boldsymbol{\Psi}$ and $\boldsymbol{\theta}$. This allows us to write

$$
\begin{aligned}
&P(c_i = k|\boldsymbol{\eta}, \boldsymbol{r}, \boldsymbol{\Psi}, \boldsymbol{\theta}, \boldsymbol{\mu}, \boldsymbol{\kappa}, \boldsymbol{x}, \boldsymbol{\omega}, \boldsymbol{\alpha}, \boldsymbol{\beta}, \lambda) \propto \\
&\begin{cases}
\frac{n_k}{n-1+\alpha_0} \prod_{j=1}^{m} P(x_{ij}|r_{jk}, \Theta) P(\Theta) & \text{if } c_i = k \\
\frac{\alpha_0}{n-1+\alpha_0} \prod_{j=1}^{m} \int P(x_{ij}|r_{j(k+1)}, \Theta) P(\Theta) P(r_{j(k+1)}) dr_j & \text{if } c_i = k+1
\end{cases}
\end{aligned} \tag{7}
$$

Here, the term before the product represents the prior $P(c_i = k)$ which Intuitively captures the cluster's proportion. The term $\Theta$ captures all other variables in the above likelihood not explicitly written again (for clarity). Since the integral over each $r_{j(k+1)}$ here is intractable, we follow Neal 2000 and sample vector $\boldsymbol{r}_{(k+1)}$ from its prior distribution when attempting to open a new cluster. We choose the prior to be $\delta_j = P(r_{jk} = 1)$ for each element for the vector $j$. Note that $\boldsymbol{\omega}$ is not included in the notation above for clarity but is trivial to include. With $\boldsymbol{\Psi}$ and $\boldsymbol{\theta}$ integrated out, we will only need to then explicitly sample $\boldsymbol{r}, \boldsymbol{\alpha}$ and $\boldsymbol{\beta}$. The full posterior conditional of $r_{jc_i}$ is given below.

$$
P(r_{jc_i} = 1|\boldsymbol{\eta}, \boldsymbol{r}, \boldsymbol{\Psi}, \boldsymbol{\theta}, \boldsymbol{\mu}, \boldsymbol{\kappa}, \boldsymbol{x}, \boldsymbol{\omega}, \boldsymbol{\alpha}, \boldsymbol{\beta}, \lambda) = \frac{P(x_{ij}|r_{jc_i} = 1, \eta_{ij}, \mu_{j1}, \mu_{j0}, \kappa_1, \kappa_0)}{\sum_{r_{jc_i}} P(x_{ij}|r_{jc_i}, \eta_{ij}, \mu_{j1}, \mu_{j0}, \kappa_1, \kappa_0)} \tag{8}
$$

However, due to high correlation between the $r_{jc_i}$s, we must sample them simultaneously. In other words, rather than sampling $r_{jc_i}$ for each $c_i = k$, sample a vector $r_j$. with length $k$. Note that as $k$ becomes large (i.e., the number of clusters grows in each MCMC iteration), computing this posterior becomes intractable since there are $2^k$ binary vectors. Therefore, we approximate this posterior by sampling $r_j$. in blocks of $r_{ja} ... r_{jb}$ where $b - a$ is the maximum blocksize.

Finally, to sample $\mu_{j0}$ and $\mu_{j1}$, we use a discrete approximation. We use possible values of $\mu$ on the interval $p \in [0.025, 0.975]$ from 20 discrete bins. Thus we can evaluate the posterior of $\mu$ using a discrete categorical distribution defined as:

$$
P(\mu_{j0} = p|\boldsymbol{r}, \boldsymbol{\Psi}, \boldsymbol{\theta}, \boldsymbol{\mu}, \boldsymbol{\kappa}, \boldsymbol{x}, \boldsymbol{\omega}, \boldsymbol{\alpha}, \boldsymbol{\beta}, \lambda) = \frac{P(x_{ij}|c_i, r_{jc_i}, \eta_{ij}, \mu_{j1}, \mu_{j0} = p, \kappa_1, \kappa_0)}{\sum_p P(x_{ij}|c_i, r_{jc_i}, \eta_{ij}, \mu_{j1}, \mu_{j0} = p, \kappa_1, \kappa_0)} \tag{9}
$$

The concentration $\kappa_0$ and $\kappa_1$ are hyperparameters that are inversely proportional to the variance of the prior distribution. We choose $\kappa_0 = 20$ and $\kappa_1 = 10$ to model low expected variance of the background and high expected variance of the signal.

**Posterior summary and convergence**

To obtain a point estimate for $r_{jk}$ and $c_i$, we apply the following posterior summary procedure. First, we obtain the pairwise matrix of probabilities that any two samples clustered together across all posterior samples (after burn-in and thinning). In other words, the probability that $c_i = c_j$ for any two samples $i$ and $j$. Determining the portion of the chain to discard can be evaluated using the Heidelberger-Welch diagnostic (Supplementary Note 3.2). However in practice, we found for real high dimensional splicing data as we used here that the estimated model parameters converge very quickly and exhibit low

posterior variance. In such cases, few MCMC iterations are needed and the optimization can be treated as a variational Bayes approximation (Supplementary Note 3.1). Convergence is then determined to be where the posterior likelihood of the model stops changing. We apply hierarchical clustering to the pairwise probability matrix with the number of clusters $k$ being the median number of clusters across all posterior samples to obtain final clustering assignments. To obtain a point estimate for $r$, we obtain a matrix of the marginal probabilities that sample $i$ in LSV $j$ is assigned to the signal distribution. $r_{jk} = 1$ if the mean of $r_{jc_i} \forall c_i = k > 0.7$. Note that we also provide an alternative approach to point summary by using the posterior sample that minimizes the MSE to the posterior mean. In other words, we generate the mean pairwise clustering matrix and pick the sample that minimizes MSE to this mean matrix.

### Reporting summary

Further information on research design is available in the Nature Portfolio Reporting Summary linked to this article.

## Data availability

The beatAML dataset can be accessed through the National Cancer Institute (NCI) at https://www.cancer.gov/about-nci/organization/ccg/blog/2019/beataml. The Therapeutically Applicable Research to Generate Effective Treatments (TARGET) dataset, phs000218, managed by the NCI can be accessed at www.ncbi.nlm.nih.gov/projects/gap/cgi-bin/study.cgi?study_id=phs000218.v22.p8. Information about TARGET can be found at http://ocg.cancer.gov/programs/target. The Penn HTSC dataset is available at GEO (GSE142514). The ENCODE knockout and eCLIP datasets from Van Nostrand et al. 2020 are available at https://www.encodeproject.org[57]. The GTEx v7 whole blood data is available at https://www.gtexportal.org/home/datasets. Source data are provided with this paper. All processed datasets are available in the Zenodo repository associated with this publication. The data generated in this study including algorithm output and data used in figures is described in the Supplementary Information and Source Data files and can be accessed in the Zenodo repository. Source data are provided with this paper.

## Code availability

All code for the algorithm, Python API and GAMBIT is publically available at https://bitbucket.org/biociphers/chessboard/src/master/. A list of Python package dependencies (pandas, scipy, numpy, seaborn, statsmodels, scikit-learn, matplotlib) are listed in the installation instructions in the repository and will be automatically installed when installing our software. The GAMBIT tool is available online at https://paros.pmacs.upenn.edu/gambit/. Sample data for GAMBIT can be downloaded from the bitbucket repository. Documentation for the CHESSBOARD python API can be found at https://chessboard.readthedocs.io/en/latest/index.html. All code to reproduce figures and analysis can be found in the Zenodo repository.

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

## Acknowledgements

We would like to thank Matthew Gazzara, Dr. Martin Carroll and Dr. Sara Cherry for their input on the data analysis interpretation, Dr. Caleb Radens for assistance with data processing, and Joseph Aicher for early input on the model/algorithm. This work was supported by NIH grants R01 GM128096 (Y.B.) and U01 CA232563/CA232563-01S3 (Y.B., A.T.T, and K.W.L.) and grants from CureSearch For Children's Cancer (A.T.T and Y.B.) and Emerson Collective (A.T.T., Project 886246066). D.W. is supported by NIH fellowship 1F31CA265218-01.

## Author contributions

Y.B. conceived the project. D.W. and Y.B. developed the methods and planned the experiments. D.W. wrote the code and carried out the experiments and analysis. M.Q.V. performed the gene ontology analysis. S.J. and M.E. developed the GAMBIT tool. M.Q.V., K.L., and A.T.T. provided feedback on data analysis interpretation. D.W. and Y.B. wrote the paper with input from M.Q.V., K.L., and A.T.T. All authors read and approved the final paper.

## Competing interests

The authors declare no competing interests.
