## [Peer Review File · Nature Communications]

A Bayesian model for unsupervised detection of RNA splicing based subtypes in cancersREVIEWER COMMENTS

Reviewer #1, expertise in bioinformatics, Bayesian models and splicing (Remarks to the Author):

The Authors present CHESSBOARD, an unsupervised algorithm for identifying subtypes of patients (and of splicing tiles) from alternative splicing data, which could be inferred from RNA-seq data. The method may particularly relevant to define tumor subtypes in heterogeneous cancers.

CHESSBOARD is based on a Bayesian model that explicitly handles i) the uncertainty in splicing quantifications and ii) missing data (by assuming they are missing not at random). The model works with "percent spliced in" event-based alternative splicing events, and inputs data on local splicing variations (LSV). Further to performing clustering, CHESSBOARD also provides a ranking of LSVs, which may enable prioritizing driver genes in downstream analyses.

The Authors benchmark CHESSBOARD in a convincing set of simulations and real data analyses.

CHESSBOARD is released as an open source Python package, and is accompanied by GAMBIT, an online visualisation tool that facilitates visual inspection of results from CHESSBOARD. CHESSBOARD is also accompanied by a clear documentation and an example usage tutorial, which greatly facilitate method's usage.

The Authors tackle a challenging problem (joint unsupervised clustering of patients and splicing tiles, from alternative splicing data) with a complex mathematical approach, and present a software tool, CHESSBOARD, which could be highly useful in clinical settings. The manuscript is clear and well written, although I feel some sections in Results are very dense, and they would be easier to follow if they had sub-headers or were split in sub-sections (if allowed by the journal).

Overall, I have a positive opinion about this work, and although I have various comments, I believe they should all be relatively easy to address.

Simone Tiberi, University of Zurich

Major comments:

1) As far as I could see, a computational benchmark is missing.

It would be important to run a computational benchmark, measuring the time and RAM usage of the method.

Also, can CHESSBOARD take advantage of multiple cores?

2) How is convergence going to be assessed when CHESSBOARD is employed by users not familiar with MCMC? Can traceplots be visualised by users?

Although the model is multi-dimensional, key traceplots, such as the overall log-posterior distribution of the model, could be used to assess convergence (e.g., via Heidelberger and Welch stationarity test).

Minor comments:

1) CHESSBOARD uses a Beta model for the percent spliced in (PSI) of each LSV.

Clearly, this would not fit gene-expression or transcript-level (relative or total) abundance.

Could the Authors please comment about the possibility of using CHESSBOARD on gene- or transcript-level data?

One trivial option (but possibly an inaccurate one) that does not require changing the current formulation, may consist in considering the ratio of reads supporting a gene or isoform.

2) Page 29: "We then select all LSVs with $p < 0.05$. This can be interpreted as selecting LSVs that are multimodal with approximately 5% being false positives."

This is actually inaccurate, as it represents the exact definition of the false discovery rate, and does not hold for raw p-values.

3) Users may be interested in employing different resolutions of clustering of patients and tiles. Does CHESSBOARD allow users to set resolution-related parameters?

As far as I understand, it seems possible indirectly via "Concentration" and "Regularization" parameters; it would be useful to explain how one can twist them to modify the resolution of clusters.

4) Page 3: claims that "event based splicing" is "the standard in the RNA splicing field".

This is certainly debatable; all 3 approaches described have advantages and disadvantages, and some may be preferred in some circumstances, but I'd certainly say event based splicing is not the "standard" approach.

5) Would it be possible to provide a measure of confidence of the clusters identified?

6) I noticed various objects are introduced without having been defined: MLE, AUC, "delta" symbol, FC, n (number of observations), p (p-value).

7) Page 16: if AUC is in [0,1], why is deltaAUC much higher (also 64)? I assume AUC is expressed in terms of percentage in [0,100] ? This should be clarified when defining AUC.

8) Page 14: at the bottom when discussing results about FLT3-ITD and NPM1, the Authors report a p-value (for comparing trees) only about FLT3-ITD ($p = 0.034$).

I would be fair to also report the p-value for the same test also for NPM1.

Also, please clarify the null and alternative hypotheses this test is comparing: is the test comparing "FLT3-ITD alone" vs. "combined classification" or combined classification vs. null/random classification?)

9) Page 10: "We trained CHESSBOARD model ...".

This train/test situation resembles a supervised learning approach, while CHESSBOARD performs unsupervised learning...I found this analysis highly confusing.

10) Page 11: "Overexpression of SRSF1 has been associated with aberrant splicing of several apoptotic factors in cancer." -> this sentence should be supported by suitable references.

11) Page 12 (mid-page): "many genes" -> this is quite vague; I'd be more specific.

12) Page 43: "by using. The signal" -> "by using the signal" (typo).

Reviewer #2, expertise in leukaemia genomics and alternative splicing (Remarks to the Author):

The authors describe Chessboard as an unsupervised method for studying alternative RNA splicing from RNA next generation sequencing data (RNA-seq) and as a new tool to identify substructures in RNA splicing data and GAMBIT as visualization tool. A number of methods already exist (reviewed in Liu, BMC Bioinformatics, 2014/Mehmood, Briefings in Bioinformatics, 2019/Muller, BMC Bioinformatics, 2021) that differ a.o. in their definition and quantification of splice events. In contrast to some other methods, the authors aim for a method that is able to deal with local splice event distributions, rare events, variability within the data and missing values. With regard to the latter, the assumption in the presented method is that values are not randomly missing. In brief, the method filters irrelevant events to reduce the data and subsequently fits a Bayesian tile model and visualizes summary statistics. They tested the tools on synthetic data (Penn-HTSC, Target) and on data sets including data with experimental drug response measurements in AML samples (BeatAML).

The process of splicing is complex and potentially powerful for cell functionality, hence the analysis of this process is highly relevant in cancer research. The tools presented in the current manuscript might be helpful to support splicing analyses and help deal with missing values in order to define subgroups of patients/cancer types with specific characteristics such as mutations or drug response. To fully appreciate the validity of the described approach more details for the bio-

informatician would be helpful while a biologist would benefit from a more thorough validation of the findings.

Specific comments:

- 1) The functional evidence that the splicing sub tiles that explain drug response are valid is not clear. And what is the reason to select the 70 genes often mutated in AML? JQ1 is shown (Fig 5b) to have a (small?) relation with the splicing cluster. Is in Fig 5d, the added value of the third analysis combining FLT3/ITD status with splicing significant?, both statistically and biologically? Are the top bars in Fig 5d an annotation of the columns from the heatmap in 5a? Where is the relation shown with NPM1 and venetoclax?
- 2) The authors list the number of junction spanning reads as an obstacle in determining Ψ . Do the authors think their method can also be successfully applied to long-read sequencing (e.g. do their model assumptions hold up)?
- 3) How did the authors reach the level of 10 reads per LSV in order to determine whether its quantifiable? What happens if you use less than 10 LSVs?
- 4) The authors perform a survival analysis that is not statistically significant (Fig. S5) but nevertheless determine "some evidence" of higher survival. Where do the authors find this evidence? Could the absence of an adverse subtype just be as likely an explanation as "missing survival data" given that the results are not significant? Please reflect upon this more elaborate in the text.
- 5) Which empirical evidence do the authors refer to when they mention "most splicing data distributions can be effectively represented as bi-modal" in the discussion?
- 6) The authors cite the possibility of validation with RT-PCR as one of the benefits for their approach compared to other methods in the introduction. Did the authors perform such analyses?
- 7) Did the authors perform any analyses to assess the accuracy of their imputation (e.g. with RT-PCR)?
- 8) Priors for background 'missingness' are estimated from non-cancer data. No details on this are provided. Is this valid? What data was used and how?

Minor comments:

- 1) A detailed algorithm visualization/scheme would be helpful for interpretation
- 2) The relatively large legend of Fig 5d is superfluous as the groups are already mentioned in the figure.
- 3) The AML tile as discovered by CHESSBOARD seems rather homogeneous which is unexpected in a disease displays a lot of heterogeneity with regard to recurrent mutations or epigenetic lesions. More details on the tile could be shown.

Responses to Reviewer Comments

We would like to thank both reviewers for their detailed and insightful comments. Addressing those helped us significantly improve the manuscript both in terms of clarity and in terms of including previously missing details/analysis. Please see below detailed responses to each of the concerns raised by the reviewers (responses are marked in blue). We believe we were able to thoroughly address the reviewers' concerns and hope you find the revised manuscript suitable for publication.

Reviewer #1, expertise in bioinformatics, Bayesian models and splicing (Remarks to the Author):

The Authors present CHESSBOARD, an unsupervised algorithm for identifying subtypes of patients (and of splicing tiles) from alternative splicing data, which could be inferred from RNA-seq data. The method may particularly relevant to define tumor subtypes in heterogeneous cancers.

CHESSBOARD is based on a Bayesian model that explicitly handles i) the uncertainty in splicing quantifications and ii) missing data (by assuming they are missing not at random).

The model works with "percent spliced in" event-based alternative splicing events, and inputs data on local splicing variations (LSV).

Further to performing clustering, CHESSBOARD also provides a ranking of LSVs, which may enable prioritizing driver genes in downstream analyses.

The Authors benchmark CHESSBOARD in a convincing set of simulations and real data analyses.

CHESSBOARD is released as an open source Python package, and is accompanied by GAMBIT, an online visualisation tool that facilitates visual inspection of results from CHESSBOARD.

CHESSBOARD is also accompanied by a clear documentation and an example usage tutorial, which greatly facilitate method's usage.

The Authors tackle a challenging problem (joint unsupervised clustering of patients and splicing tiles, from alternative splicing data) with a complex mathematical approach, and present a software tool, CHESSBOARD, which could be highly useful in clinical settings.

The manuscript is clear and well written, although I feel some sections in Results are very dense, and they would be easier to follow if they had sub-headers or were split in sub-sections (if allowed by the journal).

Overall, I have a positive opinion about this work, and although I have various comments, I believe they should all be relatively easy to address.

Simone Tiberi, University of Zurich

Major comments:

1) As far as I could see, a computational benchmark is missing.

It would be important to run a computational benchmark, measuring the time and RAM usage of the method.

Also, can CHESSBOARD take advantage of multiple cores?

We agree such analysis is important and was indeed previously missing. We have now conducted this analysis and reported the methods and results in “Supplementary Note 2.5: Runtime and Memory Evaluation”. In short, we generated synthetic data with varying parameters that affect the theoretical time complexity of the algorithm and measure the mean runtime per iteration and peak memory usage of the algorithm. Currently, CHESSBOARD cannot use multiple cores. We evaluated possible options before the initial submission of the manuscript but determined, to the best of our knowledge based on separate runtime experiments (not shown), that parallelization using these options does not help improve performance. The sample-wise likelihood computations must be performed sequentially since posteriors are updated after each sample reassignment. The feature-wise likelihood computations can be parallelized since they are computed independently per feature. However, we found that the overhead cost of creating new threads for each feature (or block of features) was exceeding the gain in runtime benefit since this step is not very computationally intensive even if done sequentially (i.e. the runtime bottleneck is in the sample-wise likelihood computations) and has to be redone on each new iteration. However, all code is open source and we would welcome future collaborations with experienced developers who may be able to optimize this process.

2) How is convergence going to be assessed when CHESSBOARD is employed by users not familiar with MCMC? Can traceplots be visualised by users?

Although the model is multi-dimensional, key traceplots, such as the overall log-posterior distribution of the model, could be used to assess convergence (e.g., via Heidelberger and Welch stationarity test).

We implemented tools for the user to visualize the log-posterior traceplots and assess convergence using the Heidelberger-Welch test. The implementation details for the test and additional experiments are described in “Supplementary Note 3.2: Convergence Diagnostics”. We have also updated our online documentation for new relevant functions for these tasks. To our knowledge, there is no existing python implementation for this test. We cross referenced the output of our implementation with the R code from packages coda, LaplaceDemon and BANDITS to check for correctness. The test statistic computations are similar however we ran into some discrepancy. Looking into when the discrepancy occurs, we believe the existing pcramer function which computes the Cramer-Von-Mises distribution CDF may have numerical instability issues because it was not computed in log space. We found that for large values of the test statistic, the CDF begins to decrease. For example, $\text{pcramer}(0.1) = 0.41$, $\text{pcramer}(4) = 0.99$. But $\text{pcramer}(100) = 0.81$. Our implementation is based on the scipy function `_cdf_cvm_inf(x)` which returns ~ 0.99 for $x > 4$. Based on our testing we think we were able to resolve the issue and adequately address the reviewer's request.

Regardless of the above, we note that as stated in “Supplementary Note 3.1: Standard CHESSBOARD Pipeline for Real Data Analysis”, on moderately high dimensional data, the algorithm in practice converges very quickly to a posterior with little to no variance. We believe it is still acceptable to run the algorithm as a variational EM method, using best approaches for running

EM algorithms (like checking multiple starting conditions). However, we note that CHESSBOARD is indeed capable of exploring robust posteriors for lower dimensional data. Specifically, we show in the new Fig S3. and Supplementary Note 3.2: Convergence Diagnostics that on small and high variance datasets, CHESSBOARD identifies clusters corresponding to high and low confidence groupings. The traceplots show robust exploration of the posterior. We have added a brief comment about this analysis in the methods under “Posterior Summary and Convergence” on page 28.

Minor comments:

1) CHESSBOARD uses a Beta model for the percent spliced in (PSI) of each LVS.

Clearly, this would not fit gene-expression or transcript-level (relative or total) abundance.

Could the Authors please comment about the possibility of using CHESSBOARD on gene- or transcript-level data?

One trivial option (but possibly an inaccurate one) that does not require changing the current formulation, may consist in considering the ratio of reads supporting a gene or isoform.

This is an excellent point for discussion. Indeed, we originally aimed to build a tile based general model that can generalize to many data modalities beyond splicing. However, analysis of gene expression read count or TPM data showed that our bimodal assumption generally does not hold well. CHESSBOARD relies on this assumption to partition each feature (LSV, gene, etc) into a background and signal group, therefore modifications to the model would be necessary. Furthermore, modeling expression this way would incur an additional computational cost due to having to model overdispersion. Using isoform ratios as the reviewer suggests is likely a viable option. As the reviewer stated, this would not require changing the current formulation as in theory, CHESSBOARD can be used to cluster any kind of data that can be expressed as ratios. However, a user should be aware of the consequences of using such an approach. For example, it may be difficult to accurately quantify isoforms using only short read RNASeq data.

2) Page 29: “We then select all LSVs with $p < 0.05$. This can be interpreted as selecting LSVs that are multimodal with approximately 5% being false positives.”

This is actually inaccurate, as it represents the exact definition of the false discovery rate, and does not hold for raw p-values.

This was an error and we thank the reviewer for pointing it out. We have corrected the statement to “This can be interpreted as selecting a LSV that is multimodal with a 5% chance of being a false positive.” The correction for FDR is discussed in the next sentence which discusses usage of Benjamini-Hochberg. Note that this does not affect our results. This is simply a filtering step and extra false positives (unimodal LSVs) will be flagged as background only by the algorithm (at the cost of a bit of runtime). We actually encourage users to use less stringent FDR/multiple testing corrections here so potentially interesting signals are not accidentally filtered.

3) Users may be interested in employing different resolutions of clustering of patients and tiles.

Does CHESSBOARD allow users to set resolution-related parameters?

As far as I understand, it seems possible indirectly via “Concentration” and “Regularization” parameters; it would be useful to explain how one can twist them to modify the resolution of clusters.

This is another good point about CHESSBOARD's usage raised by the reviewer. The reviewer is correct that in general, it is not possible to directly set the number of clusters/tiles that the user wishes to find. As the reviewer pointed out, parameters like regularization and concentration indirectly affect the number of clusters that will be found and these can be adjusted by the user using the `-A` or `-conc` flag. Regularization can be set using the `-L` or `-reg` flag. However, as we show in "Supplemental Note 2.4: Infinite Mixture Model Behavior Evaluation" the number of clusters found can significantly depend on data characteristics such as the strength of a signal as measured by the KLD between signal and background or the number of LSVs that support a tile. As a result, we can only say that the number of clusters will generally increase with increasing concentration and decrease with increasing regularization. This is stated in our documentation for the user's benefit. However, while correlated, it is impossible to predict the magnitude of the effect since it depends so heavily on data characteristics. We generally don't recommend trying to optimize the number of clusters directly as is done with classic algorithms such as k-means (e.g. using model selection heuristics on several runs with different concentrations). Users may still want to vary the concentration and check if more/less clusters are discovered. If not, the data simply does not support the presence of additional clusters given the input feature set.

Related to the question above is the recursive strategy we employ to handle this data dependent task. A finer resolution (and more clusters) can be achieved by recursively removing signals (i.e. features in a tile) and reclustering the data (Supplemental Note 4.1: Recursive Clustering and Termination). We show that this strategy is effective in finding alternate signals (Page 13).

4) Page 3: claims that "event based splicing" is "the standard in the RNA splicing field". This is certainly debatable; all 3 approaches described have advantages and disadvantages, and some may be preferred in some circumstances, but I'd certainly say event based splicing is not the "standard" approach.

While we agree with the reviewer that each of the approaches have their advantages, we respectfully still stand by our statement that event based splicing quantification is indeed the *current* standard in the RNA splicing field. We note here that this statement is about the *RNA splicing field*, not all Genomics research in general. Indeed, methods for estimating transcript abundance are widely used and are highly useful. But in the field of RNA splicing, in most cases researchers are interested in whether a condition changes the abundance of one isoform relative to another and use RT-PCR as the golden standard to validate such changes (unlike qPCR, which is considered highly inaccurate in the RNA field). With that said, we also understand the reviewer's concern. Specifically, readers may misinterpret the statement about the usage of event based quantification in the RNA splicing field as a claim against full isoform quantification. Thus, we simply removed the above statement and only described specific advantages of event based quantification in this setting (p. 3 of the main text).

5) Would it be possible to provide a measure of confidence of the clusters identified?

We provide two metrics to assess the confidence in identified clusters. The first is the probability of pairwise cluster assignment. This is represented as a sample x sample matrix where each cell of the matrix is the probability that the pair of samples were assigned to the same cluster across all MCMC iterations. The second is the marginal probability of a matrix entry being signal or background. This

is represented as a LSV x sample matrix where each cell is the probability that the corresponding cell in the data matrix was assigned to signal or background. An example of this usage is shown in Figure S2a for the beatAML analysis.

6) I noticed various objects are introduced without having been defined: MLE, AUC, “delta” symbol, FC, n (number of observations), p (p-value).

We now make sure to introduce the terms after the first instance in the text where the acronym appears.

- MLE: Page 7
- AUC: Page 15
- “delta” symbol: Page 10, Page 16
- FC: Page 16
- All instances of “n” which were used to describe the number of samples in a dataset have been change to “samples”
- p: Page 14

7) Page 16: if AUC is in [0,1], why is deltaAUC much higher (also 64)? I assume AUC is expressed in terms of percentage in [0,100] ? This should be clarified when defining AUC.

We have added substantial detail about the IC50 and AUC measurements and discussion about the limitations of the methods used in the beatAML study to obtain these measurements in Supplementary Note 5.1. In short, AUC is not constrained to [0,1], nor is it a normalized quantity. It is a unitless quantity that is only useful for making relative comparisons (a sample with a higher AUC has worse drug response than a sample with lower AUC but no further interpretation can be made). AUC itself is derived from IC50 which has the units of uM (micromolar). However, due to limitations of the beatAML study and their measurements, IC50 cannot be used for direct group-group comparisons. It can only be used to determine whether a significant difference in AUC space is biologically meaningful as indicated by a large corresponding fold change in IC50 space (> 2 generally accepted in the field). This is why we always report the change in these measurements between groups together in the manuscript but use AUC in all of our statistical testing.

8) Page 14: at the bottom when discussing results about FLT3-ITD and NPM1, the Authors report a p-value (for comparing trees) only about FLT3-ITD (p = 0.034).

I would be fair to also report the p-value for the same test also for NPM1.

Also, please clarify the null and alternative hypotheses this test is comparing: is the test comparing “FLT3-ITD alone” vs. “combined classification” or combined classification vs. null/random classification?)

We apologize for the lack of clarity. We now report the p-value for the decision for NPM1 as the reviewer requested. The details for how the permutation test was performed is in “Supplemental note 5.3: Decision Tree Permutation Test”. We have added a few clarifying statements to this section. The hypothesis is that the combined classification improves drug response prediction compared to using mutation status alone. The null is that the combined classification does not improve prediction compared to mutation status. We do not do any completely “null/random” permutations where the samples are randomly split into groups - the first layer of stratification is fixed by mutation status.

9) Page 10: “We trained CHESSBOARD model ...”.

This train/test situation resembles a supervised learning approach, while CHESSBOARD performs unsupervised learning...I found this analysis highly confusing.

Although CHESSBOARD performs unsupervised learning and does not require labeled data for its training, it still optimizes a set of latent parameters (which we described as “training”). Given the learned parameters, the model can be used to assign likelihood to unseen “test” data. For unsupervised tasks, instead of testing for accuracy (or another metric that requires labels), we evaluate performance by assessing the likelihood of unseen data either “hidden” from the same cohort (as in cross validation for labeled data) or on independent replication dataset - See Fig S4b. In addition, since the algorithm at hand is tasked with identifying tiles and assigning samples to those, we can use the trained model to assign samples to groups matching the tiles both in the training data and the test data. Then we can assess how well the $\Delta\Psi$ between the 2 assigned groups in the test data correlate with the $\Delta\Psi$ between the 2 groups discovered in the “training” dataset - see Fig 3c. Naturally, if the model does not generalize well the likelihood of the test data would not be high and accordingly the $\Delta\Psi$ found in the test data would not match well that in the training dataset. Computing the likelihood given the trained model is described in Supplemental Note 3.3 and we have added to it additional details about how the prediction was done.

10) Page 11: “Overexpression of SRSF1 has been associated with aberrant splicing of several apoptotic factors in cancer.” -> this sentence should be supported by suitable references.

Very true. We added the following two references (which we previously included elsewhere) following this sentence to improve readability/clarity:

- Das, S. & Krainer, A. R. Emerging functions of SRSF1, splicing factor and oncoprotein, in RNA metabolism and cancer. *Molecular Cancer Research* 12, 1195–1204 (2014). ISBN:1541-7786 Publisher: AACR.
- Anczuk'ow, O. et al. The splicing factor SRSF1 regulates apoptosis and proliferation to promote mammary epithelial cell transformation. *Nature structural & molecular biology* 19, 220–228 (2012). ISBN: 1545-9985 Publisher: Nature Publishing Group.

Numerous examples of SRSF1 overexpression resulting in splicing induced oncogenic behavior of target genes is given, including BIN1 and CASP9 which we subsequently analyze.

11) Page 12 (mid-page): “many genes” -> this is quite vague; I'd be more specific.

We now provide a supplementary table listing all GO terms used to draw the enrichment map shown in Fig S2D. We changed the text to better represent the meaning of the GO analysis: “The analysis shows that genes with differential splicing in the tile are enriched for roles related to general functions commonly found in cancer transcriptomics studies such as gene expression and transcription, RNA processing, and post-translational modifications (Fig. S4d).

12) Page 43: “by using. The signal” -> “by using the signal” (typo).

We have corrected this typo by removing “by using”.

Reviewer #2, expertise in leukaemia genomics and alternative splicing (Remarks to the Author):

The authors describe Chessboard as an unsupervised method for studying alternative RNA splicing from RNA next generation sequencing data (RNA-seq) and as a new tool to identify substructures in RNA splicing data and GAMBIT as visualization tool. A number of methods already exist (reviewed in Liu, BMC Bioinformatics, 2014/Mehmood, Briefings in Bioinformatics, 2019/Muller, BMC Bioinformatics, 2021) that differ a.o. in their definition and quantification of splice events. In contrast to some other methods, the authors aim for a method that is able to deal with local splice event distributions, rare events, variability within the data and missing values. With regard to the latter, the assumption in the presented method is that values are not randomly missing. In brief, the method filters irrelevant events to reduce the data and subsequently fits a Bayesian tile model and visualizes summary statistics. They tested the tools on synthetic data (Penn-HTSC, Target) and on data sets including data with experimental drug response measurements in AML samples (BeatAML). The process of splicing is complex and potentially powerful for cell functionality, hence the analysis of this process is highly relevant in cancer research. The tools presented in the current manuscript might be helpful to support splicing analyses and help deal with missing values in order to define subgroups of patients/cancer types with specific characteristics such as mutations or drug response. To fully appreciate the validity of the described approach more details for the bio-informatician would be helpful while a biologist would benefit from a more thorough validation of the findings.

Specific comments:

1) The functional evidence that the splicing sub tiles that explain drug response are valid is not clear.

We thank the reviewer for raising this point. To clarify, we do not claim that the splicing events in the tile directly cause a difference in drug response through a functional mechanism. Our main claim is the following: we observe two specific splicing events that appear to be correlated with expression levels of FLT3 (higher inclusion of the primary junction is associated with higher expression). We determined that exclusion of the primary junction in these events leads to NMD which is the mechanism by which splicing controls the expression of FLT3. The group with higher inclusion + higher expression has better response to Sorafenib, a drug given to patients with FLT3-ITD. This is “likely” (we do not claim any causality here) due to splicing creating changes (increased FLT3 expression) that are similar to the gain of function effects of FLT3-ITD in which FLT3 is constitutively active.

And what is the reason to select the 70 genes often mutated in AML?

This is another excellent question. Previous studies established the importance for AML of mutations that cause loss of function in those genes. A recent study by our group and others (Rivera et al 2021) showed that nonsense mutations in those genes underestimate loss of functions in those genes. In the case of EZH2 for example, taking into account splicing aberrations leading to NMD and

reduced protein levels basically triples the estimated fraction of patients with EZH2 LoF compared to previous estimates. Rivera et al also showed that clustering splicing changes across those 70 genes gives rise to clear groups and several candidate regulators. Given all of the above it therefore makes sense to (a) see if clustering of splicing changes in those ~70 genes using CHESSBOARD would yield similar groupings to those reported in Rivera et al (b) check whether the transcriptome wide tile finding and consequent patient grouping gives similar grouping to that based on the 70 genes, or whether these are distinct signals. We tried to clarify these aspects in the revised manuscript on page 14-15.

JQ1 is shown (Fig 5b) to have a (small?) relation with the splicing cluster.

JQ1 does indeed have an association with the splicing cluster. However, we did not analyze it further for several reasons. First, we realized that the splicing cluster had weak explanatory power for drug response (across all drugs in our data) alone and opted to combine it with mutations (Fig. 5c/d). However, we are unaware of any mutations that clinicians would use as a diagnostic to determine whether to use JQ1 for treating a patient. Thus the majority of our subsequent analysis was focused on Sorafenib which is typically given to patients with FLT3-ITD. Sorafenib itself was a high ranking drug for FLT3-ITD. For representation purposes, we could only include the top ranking drug in Fig 5b but Sorafenib was the 2nd highest ranked drug for FLT3-ITD. Since there were several FLT3 splicing events in the tile, we then further analyzed whether these splicing changes (i.e. splicing based sample grouping) could be combined with FLT3-ITD status to improve drug correlation.

Is in Fig 5d, the added value of the third analysis combining FLT3/ITD status with splicing significant?, both statistically and biologically?

This result is statistically significant. We compute a p-value for the decision tree that combines both sources of information and the details for the p-value computation are given in “Supplemental Note 5.3: Decision Tree Permutation Test”. We suspect this is potentially biologically significant because the group that was FLT3-ITD+ had splicing that we determined to mimic the gain of function of effect of FLT3-ITD. See analysis on page 16-17 and Figure 5d-e. A Wilcoxon test p-values show that differences in group means are significant as well. However such a Wilcoxon test is less informative because it does not show the significance in terms of combining information vs only FLT3-ITD status. We are primarily interested here in comparing the combine model against the mutation only model.

Are the top bars in Fig 5d an annotation of the columns from the heatmap in 5a? Where is the relation shown with NPM1 and venetoclax?

The signal/background bar (orange/red) is the same as the clustering in 5a. We have added the violin plots for NPM1 and Venetoclax in Fig. S5c and S5d. The reviewer’s comment also led us to notice another issue with our Fig. 5e where the colors on the scatter plot were flipped (the labels were correct). The signal group is yellow in all of our other plots while the background group is orange. We corrected the coloring scheme to be consistent.

2) The authors list the number of junction spanning reads as an obstacle in determining Ψ . Do the authors think their method can also be successfully applied to long-read sequencing (e.g. do their model assumptions hold up)?

Yes, long reads could be used to improve qualifications at the isoform level and thus facilitate alternate modes of study. If accurate (or corrected) long read sequencing was used to quantify isoforms, CHESSBOARD could be used by replacing LSVs with isoform ratios. The assumptions that enable us to handle uncertain quantifications (i.e. fewer reads = higher uncertainty) would still hold in this scenario. As of now, due to limited coverage and high error rate we don't view it as a practical option for cancer genomics (See also related question from Reviewer #1 above).

3) How did the authors reach the level of 10 reads per LSV in order to determine whether its quantifiable? What happens if you use less than 10 LSVs?

Excellent question. In MAJIQ's terminology there are LSV that are "reliable" i.e. you can believe they are real (and not say a mapper artifact) and LSVs that are "quantifiable" i.e. those for which it is worth the computational effort to detect significant splicing changes. The 10 reads threshold is the default for MAJIQ (Vaquero et al 2016), the method we used to process splicing data to feed CHESSBOARD. This 10 reads threshold is a result of the variance observed in read counts data for splicing, the variance observed in splicing quantification, reported phenotypic effects, and consequently what the RNA field has developed to view as significant and reproducible splicing changes. Since reads are integer counts, if a minimum of 10 reads are used, a simple ratio estimate will yield only 11 possible discrete PSI values: {0,0.1,0.2, ..., 0.9, 1.0}. This means a single read change would change our estimate by 10%. In the splicing field, a delta PSI of > 0.2 is generally well accepted as a biologically meaningful effect size (assuming statistical test is significant given replicates, e.g. in RT-PCR experiments. Some works also use 10%). If we used fewer than 10 reads, it would be harder to identify this cutoff since there are fewer possible values of PSI. In theory you could still have LSV with say less than 10 reads but a strong switch effect that would reach statistical significance under MAJIQ's Bayesian model. In practice, we find these rather rare, noisy, and less reproducible due to the stochastic nature of RNASeq and the underlying biology. In the case of CHESSBOARD, if an LSV with fewer than 10 reads was included as input to the algorithm, it actually shouldn't affect the behavior much. The model is built to handle uncertainty in PSI quantification so a LSV with a very low read count will have much lower weight compared to one with a higher read count. However, downstream interpretability would become an issue. The event might be mapped to an interesting gene domain but you wouldn't be able to report the splicing change since our uncertainty of the quantification is so high. For these reasons we kept the default MAJIQ quantifiable filter.

4) The authors perform a survival analysis that is not statistically significant (Fig. S5) but nevertheless determine "some evidence" of higher survival. Where do the authors find this evidence? Could the absence of an adverse subtype just be as likely an explanation as "missing survival data" given that the results are not significant? Please reflect upon this more elaborate in the text.

We added a new section Supplemental Note 7.1: Survival Analysis. Here, we explain how the analysis was conducted and possible issues with the statistical testing. Specifically, we used the log rank test to compute the p-value. Although this is the most commonly used test in the field, it is under-powered to detect differences in the survival distributions when the proportional hazard assumption is not true (i.e. the hazard ratio is not constant). One scenario in which this occurs is

when the survival curves diverge and then converge. Using an alternate test which simply checks whether the survival curves are different (Kolmogorov-Smirnov test, note the data is not censored here which is a main motivation to use Cox), we find the p-value is 0.054. We have also made changes to the main text in the discussion regarding our changed view. Specifically, we briefly summarize the testing issue above and state that although we observe a difference in survival rate between the groups, the effect size of that difference is 15.8%. Whether that difference has any biological significance remains unclear but we attempted to still report the analysis performed accurately.

5) Which empirical evidence do the authors refer to when they mention "most splicing data distributions can be effectively represented as bi-modal" in the discussion?

This statement was based on our general experience with such data where, given the inherent noise in splicing quantification (see discussion above), we struggle to find clear cases of three or more distinct distributions. However, given that we don't have accurate quantitative assessment we removed this statement and include the following revised text:

...Furthermore, despite the many advantages of the CHESSBOARD model, we make several modeling assumptions that could be improved upon. For example, CHESSBOARD assumes there is only a single signal distribution. In many scenarios, there can be multiple sources of heterogeneity that lead to signal distributions that are more accurately captured by a mixture of Beta distributions. We note though that in practice, given the noisy nature of splicing and its quantification from limited read counts we did not find many clear LSV cases in the data used here that would justify the additional complexity beyond a two component mixture model of signal vs. background.

6) The authors cite the possibility of validation with RT-PCR as one of the benefits for their approach compared to other methods in the introduction. Did the authors perform such analyses?

Yes. RT-PCR validations of MAJIQ quantifications of novel splice variants were successfully performed in a previously published study from our group (Rivera et al., 2021, PNAS) using some of the same samples from the Penn HTSC cohort that were analyzed here. We also showed the functional consequences of those events with reduced protein levels using Western blot. We were originally hoping to include additional validations in this study. However, because very limited biological material was available for these primary AML samples and because the RNA and protein lysates were exhausted from RT-PCR and immunoblotting experiments performed in previous studies, we were unable to conduct RT-PCR validations specifically for the LSVs reported in this manuscript.

7) Did the authors perform any analyses to assess the accuracy of their imputation (e.g. with RT-PCR)?

Earlier RT-PCR validations on some of the exact same samples from the Penn HTSC cohort that were analyzed here had confirmed MAJIQ quantifications (Rivera et al., 2021, PNAS). More extensive assessment of MAJIQ's quantification accuracy (which feeds into CHESSBOARD's

analysis performed here) by RT-PCR, including comparison to other quantification algorithms, were done in Vaquero et al 2016, Norton et al 2017, and Vaquero et al 2021.

8) Priors for background 'missingness' are estimated from non-cancer data. No details on this are provided. Is this valid? What data was used and how?

This was indeed not clearly presented in the original submission. We replaced this statement with "We then estimate priors for the missingness rates using an Empirical Bayes procedure (Methods)." The procedure is discussed extensively in the methods. We added further clarifying changes to this section on page 25-26 and provide some discussion there about validity and best practices for users.

Minor comments:

1) A detailed algorithm visualization/scheme would be helpful for interpretation

A schematic of the work flow and high level intuition behind how the algorithm works is in Fig. 1. A plate diagram (the standard for visualizing probabilistic graphical models) is in Fig. S8 with the variables in the diagram explained in "Supplementary Note 6.1: Variable Table". We would be happy to make any changes to these figures or a new figure entirely if the review is unsatisfied with these figures.

2) The relatively large legend of Fig 5d is superfluous as the groups are already mentioned in the figure.

We agree with the reviewer that the legend is too large and is mostly redundant since the group labels are shown in the violin plots. However, the green bar representing patients with normal and abnormal splicing (as defined in Fig. 5e) is not labeled by a corresponding violin plot. So we have reduced the size of the legend but kept it in the figure so the green bar is still labeled properly.

3) The AML tile as discovered by CHESSBOARD seems rather homogeneous which is unexpected in a disease displays a lot of heterogeneity with regard to recurrent mutations or epigenetic lesions. More details on the tile could be shown.

We agree with the reviewer and were also surprised by this result initially. This is why we included the analysis using our recursive clustering method. Clustering algorithms are limited in their resolution by both hyperparameters and characteristics of the data (Supplemental Note 2.4: Infinite Mixture Model Behavior). We hypothesized that the dominant signal in beatAML was simply too strong and thus the model would struggle to find finer grain patterns. Using our recursive procedure, we removed these dominant signals and found additional variation which correlated with known mutations which is expected, as the reviewer points out, due to the heterogeneity of the disease. That said, one should also keep in mind that the algorithm is built to find these common "homogenous" (once you order them accordingly) subtypes and that many other splicing variations are (a) filtered out as we describe above (b) may still remain unassigned to any tile albeit displaying high variability.

REVIEWER COMMENTS

Reviewer #1 (Remarks to the Author):

Dear Editor and Authors,
the Authors have successfully and convincingly implemented all the corrections I suggested.

I have no further comments.

I suggest the paper to be accepted.

Kind regards,
Simone Tiberi
The University of Zurich

Reviewer #2 (Remarks to the Author):

The authors have adequately responded to our questions. The only remark I still would like to make is about the survival analysis. It is well described now but the method of calculating any statistical differences here does not indicate a biological or clinical significant difference. The survival of the first 7 months after diagnosis is exactly the same and the ultimate survival after several years is also the same. Only in between a small difference could be observed, which is difficult to explain and may be related to several factors.

Reviewer #3 (Remarks to the Author): Expert in alternative splicing, bioinformatics, and biostatistics

The authors present CHESSBOARD, an unsupervised learning method for identifying splicing 'programs' that act on subclasses of a heterogeneous cancer data set, as tiles in an alternative splicing event matrix. CHESSBOARD uses a mixture Beta distribution to model and identify potentially discriminating events (LSVs) from the background, and then implements a blocked Gibbs sampling strategy coupled with the Chinese Restaurant Process to 'distribute' them into a flexible number of tiles. The method can be iteratively run to discover new tiles. Notably, the program models uncertainty in the quantification of splicing (Psi) due to coverage, by using a Beta binomial model with a higher variance for events with low coverage, as well as a Missing Not At Random (MNAR) component accounting for secondary non-coverage related effects such as those from mutation-induced NMD. Both are innovations and are shown on synthetic data sets to significantly decrease the variant estimation error and to allow to discover tiles in the presence of large dropout and missingness rates.

As applications on real data, CHESSBOARD was run and the results were thoroughly analyzed on a large acute myeloid lymphoma (AML) dataset, beatAML, and shown to identify biologically significant (sample, event) groups associated with different mechanisms, namely RBP- versus mutation-dependent, and results were then reproduced in a different AML data set, Penn-HTSC, among others.

The modeling is complex and rigorously presented and substantiated, the software -both CHESSBOARD and the visualization tool GAMBIT- is publicly available with documentation, and the application on the AML data sets is compelling for the usefulness and potential impact of the method. The information and analyses provided in response to previous reviews shed additional light on the program's requirements and use.

However, the manuscript presents CHESSBOARD as a resource, and as such, more details would be needed to ensure that the program is robust, applicable to a wider range of problems and data sets, and accessible to users, including interpretation of input, runtime choices and results.

Specific questions, and minor presentation 'bugs', are listed below.

Major:

1. a. While the application to the data sets illustrates CHESSBOARD's ability to identify signals, i.e. sensitivity, there is insufficient probing into its precision. For instance, it would be useful to run CHESSBOARD on a data set that is not expected to have tiles, such as a collection of normal samples.

b. There is little guidance on how CHESSBOARD should be run and results interpreted. For instance, the program is capable of identifying multiple tiles simultaneously, yet an iterative procedure was needed to identify more tiles in the knowingly heterogeneous AML data sets. (We also note that the LLR seems to drop significantly after the first iteration; Fig. 5b.) It would be useful to comment on why that was necessary. From a user standpoint, when is the recursive procedure warranted? What tools are available to help decide? What would be a good point to stop? The program does have a mathematically defined stop point, as reflected in Supplementary Note 4.1, but there are insufficient examples to determine whether this is reasonable in practice. Some practical guides including decision-guiding tools would be desired.

2. Events definition. There is a lack of clarity in how events are defined and represented in the matrix, and how that may impact the algorithm.

a. There is a discrepancy between Fig. 1a, which shows the supporting reads for both the inclusion and exclusion arms of a skipped exon event being included in the matrix, versus the description on page 5, which mentions only the most variable splice junction.

b. From the description, it seems that some events may be represented twice – for instance, once for each of the flanking splice junctions of a skipped exon (n.b., MAJIQ, in particular, which is used to generate the input LSVs, reports a skipped exon in multiple records). In general, there needs to be a clearer explanation, perhaps a diagram, of the types of alternative splicing events and how they are being represented as LSVs and used by the program.

c. If some events are reported through multiple entries, for instance exon skipping (2 records) versus alternative promoter (1 record), is it possible that they will skew the clustering?

3. Application to other scenarios and data sets.

a. The application to the beatAML data, reproduced on the Penn-HTSC data set, is compelling for the ability to detect functional signals in these two datasets and for the disease. However, the experiments on the combined beatAML and TARGET pediatric AML, and later on the TARGET B-ALL data, are not as conclusive or well argued. For the former, with few exceptions the resulting clusters seem to largely map by category, pediatric versus adult. While these can be due to mechanistic differences, as hypothesized, there can also be an unidentified batch effect. Moreover, neither of the two experiments was sufficiently investigated to substantiate the results. Further support would be needed, and ideally more data sets, if CHESSBOARD is to be presented as a more general resource for analyzing cancer data, beyond its highly successful application to the AML data.

b. CHESSBOARD can be used with large data sets, however, some cancer data sets may be smaller. What would be the minimum and/or the recommended range for the number of samples needed to be able to discover tiles?

Minor:

1. Page 6: "an elevated missingness rate" for the signal. This is a crucial assumption for the model, one that is driven by the potential effect of mutation-induced NMD and other phenomena. Is there any account of what the preponderance of such events is in cancer data or in the data sets under consideration? The synthetic data set used to evaluate the model had a missingness rate of 60%; how does this compare to what might be expected in real data?

2. Page 7, at the bottom: References to the 'left' and 'right' panels in Fig. 2c should be reversed,

to be consistent with the figure.

3. Page 10, reproducibility in the Penn-HTSC data set. What is the overlap, in LSV and gene sets, between the two sets of results? A Venn diagram type of calculation would be informative. How many data points are there in Fig. 3c, included in the correlation calculation?

5. Page 10. "Intersecting the tile's differentially spliced junctions with those observed as differentially spliced in ENCODE's RBP knockdown experiments implicated 17 RBPs". What were the criteria to define an RBP as being 'implicated'?

6. Page 12. The ranking of LSVs probabilistically, based on the likelihood gain, is a practical application for prioritizing events and genes. It would also be useful to have prioritization based on the effect size, related to the Psi value. Would this be possible with the information provided?

6. Page 15 top, and Fig. 5a. The text states "recovered 2 tiles", but Fig. 5a seems to indicate just 1 tile.

7. Page 19. "splice variations occur in genes that are commonly differentially mutated between pediatric and adult disease types". This statement needs a reference.

We would like to thank both original reviewers #1 and #2 for reviewing the revised manuscript and deeming it appropriate for publication.

We also thank the additional reviewer #3 who dedicated their time to provide additional feedback. These additional comments helped us further refine the manuscript and add additional information. We believe we were able to address reviewers #3 concerns and hope the revised manuscript will be deemed suitable for publication.

Reviewer #1 (Remarks to the Author):

Dear Editor and Authors, the Authors have successfully and convincingly implemented all the corrections I suggested.

I have no further comments.

I suggest the paper to be accepted.

Thank you! We are happy to have been able to address the comments and concerns raised to the reviewer's satisfaction.

Reviewer #2 (Remarks to the Author):

The authors have adequately responded to our questions. The only remark I still would like to make is about the survival analysis. It is well described now but the method of calculating any statistical differences here does not indicate a biological or clinical significant difference. The survival of the first 7 months after diagnosis is exactly the same and the ultimate survival after several years is also the same. Only in between a small difference could be observed, which is difficult to explain and may be related to several factors.

We are happy to have been able to adequately respond to the reviewer's questions and comments.

To make it clear we do not make claims about biological significance of the survival analysis we further revised the text describing this analysis to read:

"...The KS test statistic is $D=0.158$ and the KS p-value is 0.054. The test statistic can be interpreted as the largest difference in survival probability between the groups which is 15.8%. Specifically, The survival of the first seven months after diagnosis is similar and the ultimate survival after several years is also the same. Only in between these two time points some difference could be observed. While this represents a statistically significant difference in survival rate and we report it here for completeness, the biological significance of this difference is unclear and also difficult to explain."

Reviewer #3 (Remarks to the Author): Expert in alternative splicing, bioinformatics, and biostatistics

The authors present CHESSBOARD, an unsupervised learning method for identifying splicing 'programs' that act on subclasses of a heterogeneous cancer data set, as tiles in an alternative splicing event matrix. CHESSBOARD uses a mixture Beta distribution to model and identify potentially discriminating events (LSVs) from the background, and then implements a blocked Gibbs sampling strategy coupled with the Chinese Restaurant Process to 'distribute' them into a flexible number of tiles. The method can be iteratively run to discover new tiles. Notably, the program models uncertainty in the quantification of splicing (Ψ) due to coverage, by using a Beta binomial model with a higher variance for events with low coverage, as well as a Missing Not At Random (MNAR) component accounting for secondary non-coverage related effects such as those from mutation-induced NMD. Both are innovations and are shown on synthetic data sets to significantly decrease the variant estimation error and to allow to discover tiles in the presence of large dropout and missingness rates.

As applications on real data, CHESSBOARD was run and the results were thoroughly analyzed on a large acute myeloid lymphoma (AML) dataset, beatAML, and shown to identify biologically significant (sample, event) groups associated with different mechanisms, namely RBP- versus mutation-dependent, and results were then reproduced in a different AML data set, Penn-HTSC, among others.

The modeling is complex and rigorously presented and substantiated, the software -both CHESSBOARD and the visualization tool GAMBIT- is publicly available with documentation, and the application on the AML data sets is compelling for the usefulness and potential impact of the method. The information and analyses provided in response to previous reviews shed additional light on the program's requirements and use.

However, the manuscript presents CHESSBOARD as a resource, and as such, more details would be needed to ensure that the program is robust, applicable to a wider range of problems and data sets, and accessible to users, including interpretation of input, runtime choices and results. Specific questions, and minor presentation 'bugs', are listed below.

Major:

1. a. While the application to the data sets illustrates CHESSBOARD's ability to identify signals, i.e. sensitivity, there is insufficient probing into its precision. For instance, it

would be useful to run CHESSBOARD on a data set that is not expected to have tiles, such as a collection of normal samples.

We very much agree with the reviewer's comment regarding running CHESSBOARD on data without tiles, and we also believe we have already addressed this in previous analyses. We kindly ask that the reviewer reconsider the following two analyses.

1) Regarding precision: In scenarios when a signal is known to be present (e.g. in simulated data where we explicitly implant a signal), the precision of our model was assessed using a modified version of the recovery relevance score metric which was adapted to measure precision for the tile finding problem. As described in Supplemental Note 2.2, this metric is the average of a precision measure computed between each tile in a set of reference tiles (i.e. the simulated tiles) and the best matching discovered tile. The precision measure is defined as the total number of features and samples that intersect between a reference and discovered tile divided by the total number of reference tile features and samples.

2) Regarding assessing on data with "no tiles": As the reviewer correctly points out, the above method of evaluation is inadequate when no tiles are present in the data. However, creating data with no tiles is an ill defined problem because what clustering algorithms perceive as a tile depends on resolution hyperparameters and data specific properties. For example, if one were to cluster data with k-means, setting k to the number of data points would result in every data point being its own cluster even if no clusters were implanted in the data (e.g. all data points were simulated from a 2-d Gaussian with standard variance). Even though our algorithm is non-parametric and assumes an infinite number of clusters, factors such as the concentration hyperparameter and strength of signals (i.e. effect size of difference between signal and background distribution) can still impact what the algorithm "perceives" as clusters as shown in Supplemental Note 2.4. This is not a problem unique to our algorithm but rather a consequence of ambiguity associated with the solution for unsupervised learning tasks. Thus in a comparable "normal" dataset such as GTEX whole blood, there would undoubtedly still be detected tiles that result from residual confounding factors or tiles that simply exist due to random chance.

To circumvent the aforementioned problems associated with unsupervised tasks, we use a recursive clustering procedure to illustrate the effect of running CHESSBOARD on "null" data that is not expected to have tiles. Given a clustering resolution and dataset, we first isolate all features that were not assigned to tiles and recluster them. With the dominant signal removed, this data should begin to approach a null distribution with some residual or random signals. We know when we have reached this empirical null distribution once the likelihood ratio of the data vs a permuted configuration of the

data stops changing (one of the major advantages of using our model compared to others where such criteria cannot be defined). Further reclustering of the data should only result in clustering configurations of similar quality (as measured by data likelihood) to the null. In our experiments, we find that this terminal clustering solution correlates to non-splicing related factors like genetic mutations.

We have added a clarifying statement on page 13.

b. There is little guidance on how CHESSBOARD should be run and results interpreted. For instance, the program is capable of identifying multiple tiles simultaneously, yet an iterative procedure was needed to identify more tiles in the knowingly heterogeneous AML data sets. (We also note that the LLR seems to drop significantly after the first iteration; Fig. 5b.) It would be useful to comment on why that was necessary. From a user standpoint, when is the recursive procedure warranted? What tools are available to help decide? What would be a good point to stop? The program does have a mathematically defined stop point, as reflected in Supplementary Note 4.1, but there are insufficient examples to determine whether this is reasonable in practice. Some practical guides including decision-guiding tools would be desired.

We thank the reviewer for their feedback on our software. We are constantly trying to improve the user experience for our tool and strive to make it as user friendly as possible. In terms of general use case guidelines and interpretation, we have extensive documentation for the python package at <https://chessboard.readthedocs.io/en/latest/index.html>.

This includes installation, descriptions for all functions, and vignettes to show example use cases and interpretation.

Although we used a recursive clustering procedure in our analysis, this is not at all mandatory. In fact, the majority of our analysis and interpretation on all datasets was conducted on the original clustering of the data without the recursive procedure and still yielded many biologically significant insights. Determining whether this type of analysis is necessary is highly dependent on the questions that the user is trying to answer and characteristics of the data. Our recommendation is that if a user notices a preliminary clustering of the data is dominated by a very strong signal and wishes to explore alternate weaker signals, the recursive procedure would be a good idea. This is what we observed in the beatAML data and thus decided to use this procedure. In contrast, the pediatric AML and B-ALL datasets had a more diverse set of signals, indicating that our algorithm can pick up on more complex tile structures without the recursive procedure if sufficiently supported by the data.

An important point we would like to make is that the recursive procedure and reasons for using it are not unique to our algorithm. The concept and application are both widely used in many clustering applications in the bioinformatics field and should not require extensive justification. For example, a single cell clustering analysis of whole blood tissue might observe a population of B and T cells. But if the analyst wanted to study subtypes of T cells, they could isolate the T cells and recluster them. We of course have extensively adapted this concept for tile finding and added mathematical termination guidelines to make it more rigorous. Details for the recursive procedure can be found in Supplemental Note 4.1. Jupyter notebooks showing this analysis which can be used as a user reference will be released in a repository upon publication. Our termination criteria can be interpreted as a permutation based model selection method. This concept of assessing whether the ratio of some statistic computed on the data vs a null model comes from a null distribution is a standard statistical technique and we thus did not find it needed to justify it further.

2. Events definition. There is a lack of clarity in how events are defined and represented in the matrix, and how that may impact the algorithm.

a. There is a discrepancy between Fig. 1a, which shows the supporting reads for both the inclusion and exclusion arms of a skipped exon event being included in the matrix, versus the description on page 5, which mentions only the most variable splice junction.

We apologize for the confusion this caused and added clarifying statements to the text. Specifically, the inclusion and exclusion reads are both included as input to the algorithm as described on page 5 and as shown in Fig. 1a. Each row of the input matrix represents a splicing event represented as a LSV in MAJIQ terms. A LSV is defined with respect to an exon and includes all junctions with reads that overlap the reference exon and possible downstream (source LSV) or upstream (target LSV) exons it is spliced to. LSVs can be complex and contain more than 2 junctions. From a modeling perspective, we are interested in whether a single junction in the LSV differs between the signal and background distributions. The representative junction we choose for the model is the most variable in the LSV as measured by empirical variance. The reads assigned to all other junctions are summed to represent all possible alternative junctions. We have added a new Supplemental Note 1.2 to describe this in more detail.

b. From the description, it seems that some events may be represented twice – for instance, once for each of the flanking splice junctions of a skipped exon (n.b., MAJIQ, in particular, which is used to generate the input LSVs, reports a skipped exon in multiple records). In general, there needs to be a clearer explanation, perhaps a

diagram, of the types of alternative splicing events and how they are being represented as LSVs and used by the program.

This is a good point to clarify. First, we should point out that CHESSBOARD does **not** require MAJIQ output to function. A user may choose to supply two TSV files where each row is a splicing event of interest defined using whatever tool the user desires. The entries in the first file represent reads mapped to the junction of interest while the entries in the second file represent the sum of all reads not mapped to this junction of interest but are still contained in the splicing event/normalization unit. If one was to use rMATS for example, the user could report inclusion reads vs the average exclusion reads per rMATS events. In general, the biological model of the splicing unit is independent of the statistical model in terms of its power to detect tiles since all splicing models can be abstracted as ratios of reads. We have added more information about events types in the new supplementary note.

In our experience MAJIQ carries great advantages in its ability to detect and quantify both complex and unannotated splicing variations, both are key for cancer data. That said, the reviewer is correct that when using MAJIQ one needs to decide how to process/represent the LSVs captured by MAJIQ. As we discuss in the main text, we select LSVs based on the most variable junction and the TSV files discussed above capture the estimated read rates for that junction compared to all other competing junctions in that LSVs. In Rivera et al PNAS 2021 we specifically filtered any LSVs that overlap (share junctions) and in MAJIQ V2 (Vaquero et al BioRxiv 2021) we released the modularizer that can break gene splice graphs into disjoint areas or splicing modules. In the analysis we performed here we did not employ the modularizer (was still under development at the time), allowed overlap of LSVs if they both passed the filter yet for some reason still did not result in the same junction being selected (note that cassette single source/target LSVs would assign the same variance to the two junctions while the exclusion junction is shared), and we did not find a change in grouping compared to Rivera et al.

c. If some events are reported through multiple entries, for instance exon skipping (2 records) versus alternative promoter (1 record), is it possible that they will skew the clustering?

This is an excellent question. In principle, it is conceivable that some duplication occurs and affects clustering - See also detailed explanation above. However, we note that even if duplication does occur the impact should be small - the same underlying change can be duplicated at most twice and it is unlikely that only the junctions that support a very different tile structure will be the only ones duplicated resulting in a signal dominated by those junctions. In practice we found our filtering of LSVs and those of

Rivera et al that explicitly removed any possible overlap were extremely similar for both BeatAML and PennHTC data in terms of patient grouping though CHESSBOARD was able to detect signals missed by Rivera et al due to missingness as we discuss in the main text.

3. Application to other scenarios and data sets.

a. The application to the beatAML data, reproduced on the Penn-HTSC data set, is compelling for the ability to detect functional signals in these two datasets and for the disease. However, the experiments on the combined beatAML and TARGET pediatric AML, and later on the TARGET B-ALL data, are not as conclusive or well argued. For the former, with few exceptions the resulting clusters seem to largely map by category, pediatric versus adult. While these can be due to mechanistic differences, as hypothesized, there can also be an unidentified batch effect. Moreover, neither of the two experiments was sufficiently investigated to substantiate the results. Further support would be needed, and ideally more data sets, if CHESSBOARD is to be presented as a more general resource for analyzing cancer data, beyond its highly successful application to the AML data.

The primary purpose of the analysis on pediatric AML and B-ALL data was to show that CHESSBOARD could find more complex tile structures. As we explained in our response to section 1 of this review, the beatAML dataset tiles were primarily driven by a very strong signal that only resulted in a single tile. Here we wanted to show that our algorithm could detect more complex structures in scenarios where the data supported them without using a recursive clustering procedure. A more extensive analysis of this data at the level of our beatAML analysis is beyond the scope of this method-focused paper. We do have ongoing projects with upcoming publications that will focus more extensively on extracting more substantiated biological insights from these datasets. Consequently, we don't make any definitive biological claims about our results on these datasets besides stating several observations and hypotheses about what the clusters could potentially represent. However, in terms of CHESSBOARD's robustness to applications on other leukemia datasets, we don't think there is a compelling reason to believe our modeling assumptions for splicing would be different for these diseases.

b. CHESSBOARD can be used with large data sets, however, some cancer data sets may be smaller. What would be the minimum and/or the recommended range for the number of samples needed to be able to discover tiles?

This is another great question. As we showed in our convergence analysis in Supplemental Note 3.2 and Fig. S2, confidence in clusters goes down with decreasing dataset size. However, the Bayesian nature of our algorithm enables us to fully quantify

the uncertainty of tile assignments and make informed decisions about how to interpret the quality of the algorithm's output. If a dataset is too small to produce a high quality clustering, the output of CHESSBOARD will reflect that through low marginal probabilities of tile assignment which can be quantified and visualized using our package. In general, it is difficult to specify a definitive range of appropriate dataset sizes. If a signal is strong enough (i.e. large effect size), it can be detected using only a few samples. Conversely, a large number of samples would be required for the algorithm to have sufficient power to detect weak signals. Regardless, the tools are in place for a user to evaluate whether the algorithm's output is reasonable given a small dataset as input.

Minor:

1. Page 6: "an elevated missingness rate" for the signal. This is a crucial assumption for the model, one that is driven by the potential effect of mutation-induced NMD and other phenomena. Is there any account of what the preponderance of such events is in cancer data or in the data sets under consideration? The synthetic data set used to evaluate the model had a missingness rate of 60%; how does this compare to what might be expected in real data?

In practice, a relatively few number of events have a significant difference in missingness rate between signal and background groups (Table S1 and S3). When there is no significant difference, missing values are interpreted as dropouts. In general, clustering will be dominated by observed splicing events. However, as shown in Table S1 and S3, we report the p-values (Fisher's exact test) for difference in missingness rate between the signal and background distributions as part of the algorithm's output. When this p-value is significant, it is likely that there are non-technical factors (e.g. mutation-induced NMD) causing this difference in missingness rate. We explore potential mechanisms for several events with significant differences in missingness rate in EZH2 with PTC introducing events which were experimentally verified to affect protein level in Rivera et al. 2021.

2. Page 7, at the bottom: References to the 'left' and 'right' panels in Fig. 2c should be reversed, to be consistent with the figure.

We thank the reviewer for pointing out this error. We have corrected it.

3. Page 10, reproducibility in the Penn-HTSC data set. What is the overlap, in LSV and gene sets, between the two sets of results? A Venn diagram type of calculation would be informative. How many data points are there in Fig. 3c, included in the correlation calculation?

Both matrices have the same LSVs and thus the same gene sets (LSVs = 2299). The filtering procedure described in Methods was applied to beatAML and the same LSVs were selected in Penn HTSC. When using the model trained on beatAML to predict clusters in PennHTSC, the likelihood of each sample was evaluated under each of the 2 clusters in the beatAML model. The left cluster model had 1910 LSVs in the signal while the right cluster model had all LSVs in the background. Each sample in PennHTSC was assigned to the cluster with the highest likelihood. Fig. S4b shows that the likelihood distributions between beatAML samples and HTSC samples were similar.

5. Page 10. “Intersecting the tile’s differentially spliced junctions with those observed as differentially spliced in ENCODE’s RBP knockdown experiments implicated 17 RBPs”. What were the criteria to define an RBP as being ‘implicated’?

These were the RBPs that were either differentially expressed or spliced between the two groups (tile and background). More details about this criteria are given in Supplementary Note 3.3.

6. Page 12. The ranking of LSVs probabilistically, based on the likelihood gain, is a practical application for prioritizing events and genes. It would also be useful to have prioritization based on the effect size, related to the Psi value. Would this be possible with the information provided?

This is definitely possible but could be accomplished in many different ways, each with different interpretations. The simplest method would be to report the difference in mean or median PSI between the signal and background distributions which would be trivial to do given the algorithm’s output of cluster identities. This is analogous to what is done in differential splicing/expression studies. However, this method does not consider the variance of PSI distributions. Our likelihood based approach would accomplish this. In fact, the likelihood approach is very similar to simply reporting the test statistic of a parametric differential splicing test. However, since we are interested in rankings, we don’t necessarily need to attach a p-value to this test statistic. If a user was interested in obtaining a p-values for a test in which the null hypothesis is that the mean difference between the signal and background groups is 0, they could run a DS algorithm like MAJIQ or simply use a rank sum test given the cluster labels reported by the algorithm and the computed PSI values.

6. Page 15 top, and Fig. 5a. The text states “recovered 2 tiles”, but Fig. 5a seems to indicate just 1 tile.

We thank the reviewer for pointing out this error. We have corrected it to “recovered 2 clusters”.

7. Page 19. “splice variations occur in genes that are commonly differentially mutated between pediatric and adult disease types”. This statement needs a reference.

We have added the following 2 citations:

Ma, X., Liu, Y. U., Liu, Y., Alexandrov, L. B., Edmonson, M. N., Gawad, C., ... & Zhang, J. (2018). Pan-cancer genome and transcriptome analyses of 1,699 paediatric leukaemias and solid tumours. *Nature*, 555(7696), 371-376.

Gröbner, S. N., Worst, B. C., Weischenfeldt, J., Buchhalter, I., Kleinheinz, K., Rudneva, V. A., ... & Pfister, S. M. (2018). The landscape of genomic alterations across childhood cancers. *Nature*, 555(7696), 321-327.

REVIEWERS' COMMENTS

Reviewer #2 (Remarks to the Author):

Thank you for the adjustment and I have no further comments.

Reviewer #3 (Remarks to the Author):

All of my comments were addressed to my satisfaction in the revised manuscript and supplement. I thank the authors for their very informative answers.